



# Water and Energy budgets over hydrological basins on short and long timescales.

Samantha Petch[1], Bo Dong[1,2], Tristan Quaife[1,2], Rob King[3], and Keith Haines[1,2]

[1]Department of Meteorology, University of Reading, Reading, UK
[2]National Centre for Earth Observation, University of Reading, Reading, UK
[3]Met Office, Exeter, UK

**Correspondence:** Samantha Petch (s.petch@pgr.reading.ac.uk)

**Abstract.**

Quantifying regional water and energy fluxes much more accurately from observations is essential for improving climate and earth system models, and their ability to simulate future change. This study uses satellite observations to produce monthly flux estimates for each component of the terrestrial water and energy budget over selected large river basins from 2002 to 2013. Prior to optimisation the water budget residuals vary between 1.5 % and 35 % of precipitation by basin, and the imbalance between the net radiation and the corresponding turbulent heat fluxes ranges between $\pm$ 10 $\mathrm{Wm^{-2}}$ in the long term average. In order to further assess these imbalances, a flux-inferred surface storage (FIS) is used for both water and energy, based on integrating the flux observations. This exposes mismatches in seasonal water storage as well as important interannual variability.

Our optimisation ensures the flux estimates are consistent with total water storage changes from GRACE on short (monthly) and longer timescales, while also balancing a coupled long term energy budget, by using a sequential approach. All the flux adjustments made during the optimisation are small and within uncertainty estimates using a $\chi^2$ test, and interannual variability from observations is retained. The optimisation also reduces formal uncertainties on individual flux components. When compared with results from previous literature in basins such as the Mississippi, Congo and Huang He river, the FIS metrics show the better agreement with GRACE variability and trends in each case.

## 1 Introduction

The terrestrial water cycle largely determines the Earth's climate and causes much of the natural climate variability. Variations and long-term changes to the water cycle can have profound impacts on regional agriculture, ecosystems and society. The surface energy budget is a key driver of the global water cycle, as well as having large influence over atmosphere and ocean dynamics and a variety of surface processes. Despite the fundamental importance to our understanding of climate and climate change, there remain some key challenges to quantifying the regional water and energy cycling rates. In particular observations of the flux and storage terms tend to have large uncertainties and are inconsistent with budget considerations, while model estimates are internally consistent but usually show some mismatches to observations (e.g., Dong et al., 2020).



Water is a conservative quantity, and so, the mass balance described by the water budget in Eq. (1), should be satisfied

$$P - E - Q = dS. \tag{1}$$

This states that the precipitation (P) falling over an area, combined with the loss of water to the atmosphere through evapo-transpiration (E), and the horizontal loss of water through runoff (Q), is balanced by the change of water storage (dS) in the area. The availability of GRACE satellite gravitational measurements of total water storage (S) since 2002 (Tapley et al., 2019) has provided a valuable constraint to aid understanding of the other water budget components. Previous literature has used the budget equation to test the accuracy of observations (Reeves Eyre and Zeng, 2021) and to validate model estimates (Long et al.,

2015). Several studies also exist which use the budget equation to estimate one component using observations for the other terms (Chen et al., 2020; Wang et al., 2015; Sheffield et al., 2009). For instance, Chen et al. (2020) provide a new estimate of seasonal and yearly river runoff changes for the Amazon basin using the water budget closure method, and Rodell et al. (2011) use budget closure to estimate evapotranspiration.

Recent developments in satellite retrievals has meant that budget closure can be assessed purely from remotely-sensed

data sources (Sheffield et al., 2009). However, water fluxes are still affected by considerable uncertainties, which has been highlighted in many water budget studies when independent products are combined. For example, Sheffield et al. (2009) used the budget equation taking Q as a reference variable and found significant errors that were larger than the observed Q taken from in situ measurements. A common approach among previous water budget assessments is to use a range of products for each flux component and evaluate the ability of different combinations to close the water budget. For example, Lehmann et al. (2022)

investigated budget closure at catchment scales using 11 precipitation, 14 evapotranspiration and 11 runoff datasets, together with GRACE. The study concluded that no one combination of data sources can close the budget well for all regions. It was also highlighted that regions where selected data sources did close the budget reasonably well, could be as a result of cancellation of errors. The multi-source strategy has the potential to compensate for the limitations of each individual estimation method in terms of its accuracy, spatial and temporal coverage. However, combining multiple sources can introduce a new challenge of

how to allow for discrepancies between the different data products. Lehmann et al. (2022) determined uncertainties for each flux based on the inter-product spreads, which is the common way to treat uncertainties when multiple products have been used (e.g., Abolafia-Rosenzweig et al., 2021). Resolving the uncertainty among the various estimates for a specific variable remains an underlying challenge in using both in situ measurements and remote sensing observations (Pan et al., 2012).

Non-closure errors can come from the complexities in deriving energy and water fluxes from remote measurements. This

process involves independent algorithms that use distinct observations and assumptions which can be subject to both random and systematic errors (L'Ecuyer et al., 2015). Since each flux dataset may be developed in isolation, valuable energy budget and water cycle closure information is lost. Then reintroducing budget closure as a constraint may help to reduce biases is these datasets.

Several studies exist which produce best estimates for all components in order to close the budget (e.g., Sahoo et al., 2011).

Different techniques are seen to impose closure constraints; such as Kalman-filters (Pan et al., 2012; Zhang et al., 2018), post-filtering (Munier et al., 2014; Aires, 2014), and variational methods (Rodell et al., 2015; Hobeichi et al., 2020). Abolafia-



Rosenzweig et al. (2021) focused on human impacts on the water cycle and produced a remotely sensed ensemble of the terrestrial water budget (REESEN) containing 60 unique realisations of the water budget for basins between 50° S and 50° N, over Oct 2002-Dec 2014. Three different closure techniques were applied to all ensemble members in order to produce three

ensembles of "corrected" budget estimates. Zhang et al. (2018) produced a climate data record (CDR) for the period 1984-2010 which provides monthly 0.5° resolution global estimates of each flux component while closing the budget using a constrained Kalman-filter.

Typically, these methods produce new monthly estimates for each flux by adjusting input observations according to defined errors in order to achieve complete budget closure (Aires, 2014), or to achieve a budget residual within allowed errors (Hobeichi

et al., 2020). Errors are often based on inter-product spread Abolafia-Rosenzweig et al. (2021) or based on discrepancies with non-satellite data (Sahoo et al., 2011). Crude approximations are also sometimes used when representing errors, for example, Munier et al. (2014) supposed constant errors for P and E of 10 cm, and 10 % of the mean discharge for Q. Such assumptions are made due to the absence of any comprehensive study that quantifies errors at the global or regional scales for each of the datasets used (Munier and Aires, 2018).

Zhang et al. (2018) adjusts fluxes according to the deviation from the ensemble mean of all data sources for the same budget variable. In a post hoc adjustment, (Zhang et al., 2018) also remove an implied long-term storage trend by redistributing the non zero mean dS between the precipitation and the evapotranspiration in a way that maintains budget closure. However, most other studies that close the water budget on a monthly timescale fail to consider total water storage over longer timescales. The GRACE timeseries does provide information on water storage on all timescales longer than 1 month, and so when only using

a monthly dS as input information can be lost. Post hoc detrending (Zhang et al., 2018) is also not correct for regions where GRACE does detect a trend in storage, e.g. Wang et al. (2015) found significant trends in water storage in 11 out of the 19 basins studied. In addition GRACE data often reveal interesting interannual variations in basin water storage which will not necessarily be reproduced by most previous approaches (examples will be shown later). One key aim of this study will be to produce balanced water budgets which agree with the inter-annual variability and long-term trends observed by GRACE. Since

the other fluxes P, E and Q are linked to dS via Eq. (1), the use of the additional information given by GRACE storage should also provide more accurate constraints on these fluxes. To the authors' knowledge, no previous efforts have been made to fully fit estimates to GRACE during the optimisation closure.

The surface energy balance can be described by the incoming energy from downwelling shortwave and longwave radiation (DSR and DLR respectively), and the outgoing energy from the longwave flux (ULW), reflected shortwave flux (USW) and the

turbulent heat fluxes latent and sensible heat (LE and SH respectively). Fluxes are taken to be positive when directed towards the surface, therefore, the energy budget can be written as

$$DSR + DLR - USW - ULW - LE - SH = NET \qquad (2)$$

where NET is the total energy absorbed by the surface. The water and energy cycles are coupled due to the exchanges of latent heat that occur during precipitation and evapotranspiration and so we will also include a regional coupled energy budget

closure in our analysis, with a particular focus on seasonal to interannual variability, and the interactions with the water cycle.



Without limitations on water availability, evaporation increases with increasing temperature which must be balanced by an increase in precipitation. Additionally, warmer air can hold more moisture, about 7 % more water vapour for each degree Celsius of warming, and so, evaporation and precipitation are projected to intensify as consequence to changes in the Earths energy balance (IPCC, 2013).

The coupling between the water and energy budgets enables them to provide constraints on one another, however, most previous studies have performed water or energy budget analyses independently. The NASA Energy and Water cycle Study (NEWS) derived an optimised coupled global-continental scale budget with Rodell et al. (2015) focusing on water and the parallel energy budget L'Ecuyer et al. (2015), for the period 2000–2010 focusing on satellite derived data as far as possible. Thomas et al. (2020) extended the NEWS coupled approach focusing on improving ocean basin fluxes. Hobeichi et al. (2020)

then developed a regional coupled approach over land areas producing the Conserving Land–Atmosphere Synthesis Suite (CLASS) which solves for monthly water and energy budgets at 0.5 ° grid scale over Land areas from 2003–2009. Data-driven global flux estimates are subject to uncertainty due to the lack of energy balance closure. In order to mitigate this, some data sources look to account for energy balance within their products. For example, FluxCOM products undergo three different energy balance closure corrections for LE and SH (Jung et al., 2019).

This study produces optimised estimates for each of the the water and energy budget components, based on observations. It aims to ensure that the estimates are consistent with storage trends seen by GRACE on short and long timescales, and that the total energy lost or absorbed by the ground over this time period is small. The estimates are also constrained to close the monthly water budget whilst accounting for the uncertainties of the observations. The paper is organised as followed, Sect. 2 describes the data used as input for the optimisation, Sect. 3 describes the methodology used in the study, results are shown in

Sect. 4. Optimisation uncertainties are included in Sect. 5 and a discussion is included in Sect. 6, before concluding in Sect. 7.

## 2 Data

Each of the data sets described in this section has a monthly resolution and has been interpolated at a 0.5° spatial resolution, and then masked and spatially averaged over different basins chosen in this study. Flux data sets represent the average flux over each calendar month and therefore is considered to represent the flux mid-month. The input data sources used are summarised

in Table 1

### 2.1 GRACE

Water storage data is taken from The Gravity Recovery and Climate Experiment (GRACE). GRACE measures changes in the Earths gravity field, which is directly correlated to the change in surface mass and is indicative of water storage change. The water mass anomalies are expressed in terms of equivalent water thickness and represent the deviations of mass in terms of

vertical extent of water in centimeters. All water storage compartments including snow, surface water, soil moisture, and deep groundwater are accounted for (GravIS, 2021).

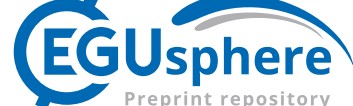

| Data source | Parameter | period resolution | spatial resolution | key references |
|---|---|---|---|---|
| CERES EBAF_Ed4.1. | Radiative fluxes | 1997-present | 1 ° | Wielicki et al. (1996) |
| GRACE-JPL MAS-CON CRIv01 | Water storage | 2002-present | 0.5 ° | Wiese et al. (2016) |
| FluxCOM | Sensible heat Latent heat | 2001-2015 | 0.5 ° | Jung et al. (2019) |
| GPCPv2.3 | Precipitation | 1979-present | 2.5 ° | Adler et al. (2003) |
| GRUNv1.0. | Runoff | 1902-2014 | 0.5 ° | Ghiggi et al. (2019) |

**Table 1.** Data sources

The GRACE data were processed using an advanced mass concentration ("mascon") approach that enables improved signal resolution relative to the standard spherical-harmonic technique (Rodell et al., 2018). It acts across coarse spatial and temporal scales and requires filtering prior to being interpreted. The processing chain of GRACE data involves a large number of corrections and uncertainties that introduce errors and impose restrictions on its use (Swenson and Wahr, 2006). One of the most important errors is the signal leakage between neighboring grid cells caused by the truncation of spherical harmonics and Gaussian filtering (Landerer and Swenson, 2012). The version used here is the MASCON JPL RL06_v2, which uses a Coastline Resolution Improvement (CRI) filter applied to separate the land and ocean portions of mass within each land/ocean mascon in a post-processing step. The relative magnitude of ocean and land leakage errors is primarily a function of how the mascon placement conforms to the coastline. The CRI filter acts to reduces leakage errors across coastlines. Wiese et al. (2016) quantifies the associated errors in determining mass variations for different basins. On average, measurement errors dominate leakage errors (mean error of 8.3 mm versus 5.1 mm), particularly for larger basins, as the fraction of fully contained mascons in the basins increases.

The data is provided with 0.5° resolution grids and time given as days since 2002-01-01T00:00:00Z. Storage values are provided per calendar month but these must be converted into storage changes, dS, over each month. In the literature, several different methods have been used such as centred difference schemes (Zhang et al., 2018), backwards difference schemes (Hobeichi et al., 2020) and fourth difference schemes (Reeves Eyre and Zeng, 2021). Here we use simple centred differences, $dS(i) = (S(i+1) - S(i-1))/2$ for month $i$.

There were a small number of months of missing data which were filled with monthly climatology plus temporal interpolation of monthly storage anomalies.

## 2.2 GPCPv2.3

Precipitation data is taken from the Global Precipitation Climatology Project (GPCP) version 2.3, see Adler et al. (2016). GPCP provides monthly precipitation data from 1979-present and aims to provide a globally coherent dataset of precipitation (Adler et al., 2003). It combines observations and satellite precipitation data into 2.5 ° global grids. The product employs precipitation





estimates from the 0600 and 1800 low-orbit satellite Special Sensor Microwave Imager (SSM/I) and Special Sensor Microwave
Imager and Sounder (SSMIS) passive microwave data to perform a calibration of infrared data from the international collection
of geostationary satellites in the latitude band 40° N-40° S. The satellites include NOAA's Geostationary Operational Environmental Satellites (GOES) and the calibration varies by month and location (Adler et al., 2016). Absolute magnitudes are
considered reliable and inter-annual changes are robust. Precipitation may be underestimated in mountainous area, however,
version 2.3 has improved on this compared to previous versions (Adler et al., 2016).

## 2.3 GRUNv1

Runoff data is taken from the GRUN dataset. GRUN provides a global gridded reconstruction of monthly runoff covering the
period 1902-2014 at a 0.5° spatial resolution (Ghiggi et al., 2019). The dataset uses a global collection of in situ streamflow
observation to train a machine learning algorithm that predicts monthly runoff rates based on antecedent precipitation and
temperature from an atmospheric reanalysis. The precipitation and temperature data are obtained from the Global Soil Wetness Project Phase 3 (GSWP3) dataset version 1.05 (Kim and Oki, 2017). The in situ runoff observations are derived from
the Global Streamflow Indicies and Metadata Archive (GSIM) (Do et al., 2018), which consists of 35002 streamflow stations.
Model validation is based on cross-validation experiments using datasets such as the Global Runoff Data Centre (GDRC) Reference Dataset (GRDC, 2020) and runoff simulations from the Inter-Sectoral Impact Model Intercomparison Project (ISIMIP)
(Warszawski et al., 2014). Different metrics are used to assess the skill of the runoff reconstruction. For large GRDC river
basins, the relative bias (which has an optimal value of 0) had a median of 0.047, the squared correlation coefficient, $R^2$, had
a median of 0.738 and the ratio of standard deviations (optimal value of 1) had a median of 1.004. Overall, the agreement is
said to be satisfactory, although there is a tendency to underestimate runoff rates when the magnitude increases (Ghiggi et al.,
2019).

## 2.4 FluxCOM

Latent and sensible heat data come from FLUXCOM using the RS+METEO set up. FLUXCOM uses machine learning to
merge energy flux measurements from FLUXNET eddy covariance towers with remote sensing and meteorological data to
estimate net radiation, latent and sensible heat and their uncertainties. Using three different machine learning algorithms, energy balance closure correction constraints, and climate forcing data from various sources as predictors, a large ensemble of
gridded flux products are generated (FluxCom, 2021). A lack of energy balance closure of around 20 % was observed across
FLUXNET sites, which was addressed using three different approaches based on hypotheses regarding primary cause of the
energy balance closure gap. Closure corrections include the Bowen ratio correction, which assumes that the ratio of sensible
and latent heat is accurately measured, and a "residual approach" which reallocates missing energy to other flux components
(Jung et al., 2019). The data is provided on 0.5° global grids. FLUXCOM ensemble products provide uncertainties per grid
cell and time step. Uncertainties can arise from empirical upscaling, the choice of machine learning algorithm and the predictor
variables.



## 2.5   CERES

This study takes radiative flux data from the Clouds and Earth Radiative Energy System (CERES); a multi-satellite measure-
ment program for monitoring radiation. CERES instruments were designed to provide accurate measurements for the long-term
monitoring of Earth's reflected shortwave and emitted longwave radiances as part of its radiation energy budget (Loeb et al.,
2016). Seven CERES instruments on five satellites have been launched (TRMM, Terra, Aqua, S-NPP, NOAA-20). Each CERES
instrument has three channels: a shortwave channel to measure reflected sunlight, a longwave channel to measure earth-emitted
thermal radiation in the 8 to 12 μm "window" region, and a total channel to measure all wavelengths of radiation. Onboard
calibration sources include a solar diffuser, a tungsten lamp system with a stability monitor, and a pair of blackbodies that can
be controlled at different temperatures (Wielicki et al., 1996). The CERES record is highly stable and has twice the spatial res-
olution and improved instrument calibration compared to the ERBE record (Acker et al., 2014). Here we use the latest version
CERES EBAF Ed4.1. This version uses new clear-sky fluxes determined for the total region to determine TOA and surface
Cloud Radiative Effects (CREs). Uncertainties are primarily determined by comparing EBAF surface fluxes with observations
at surface sites over land and buoys over ocean (Kato et al., 2018).

## 2.6   Initial Uncertainties

Many previous water budget studies have dealt with uncertainties by solving for multiple data products for each component
and using the spreads as a measure of uncertainty. We have only used single data products here although the uncertainties
applied are based on prior studies that have taken multiple products to estimate uncertainties, in particular the NEWS analysis
(L'Ecuyer et al., 2015; Rodell et al., 2015). Product uncertainties are very hard to estimate on regional scales because of
unknown spatial error covariances. In situ calibration based errors may be correlated on small spatial scales but are likely to
be uncorrelated on larger spatial scales. In addition many product errors may scale with flux amplitudes, and some previous
studies have therefore assigned uncertainties as a % of flux amplitudes.

Here, in order to give a traceable method, we have taken the continental scale uncertainty estimates from the NEWS papers
above and downscaled them to river basin scales while assuming that errors are uncorrelated between river basin scales and
continental scales. This leads to the following relationship between basin scale and continental scale flux uncertainties;

$$\sigma_f = \sqrt{(f/F).(A/a)}.\Sigma_F \tag{3}$$

where $\sigma_f$ is the basin scale uncertainty on flux $f$ over basin area $a$, and $\Sigma_F$ is the continental scale uncertainty on flux $F$
over continental area $A$. If the errors were assumed to correlate between scales the simpler uncertainty scaling as a % of flux
amplitudes would apply.

However GRACE does not measure a flux but rather the strength of the gravitational field anomaly. To calculate the dS
uncertainties we use the basin values proposed by Wiese et al. (2016) for the JPL RL05M GRACE Mascon solution. Their
method combines measurement uncertainty ($\epsilon_m$) and leakage uncertainty ($\epsilon_l$) to produce an uncertainty for storage (S). We
then calculate uncertainty in storage change between any two months (dS), assuming errors are uncorrelated from one month





to another;

$$\sigma_S = \sqrt{\epsilon_m^2 + \epsilon_l^2}, \;\; \sigma_{dS} = \sqrt{2}\sigma_S. \tag{4}$$

We show examples of both input and optimised uncertainties later in the paper.

## 2.7 Study area and period

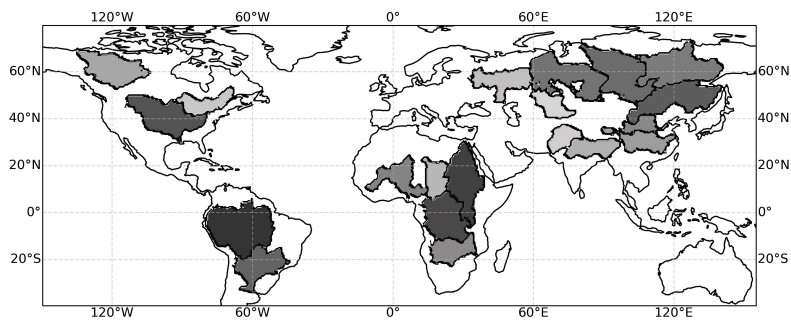

**Figure 1.** Location of 20 selected large basins.

For this study we focus on large river basins. The Mississippi, the Amazon, the Yellow River, the Congo and the Amur are selected for more detailed analyses in this paper, based on storage trends seen in GRACE and overlapping regions with other studies to enable comparisons. Additional results for a larger range of basins are shown in the Appendix. This study is carried out for 2002-2013 due to the availability of selected data products.

## 3 Methods

### 3.1 Inferred storage from observations

For both water and energy budgets we find it useful to calculate a surface storage which we call the flux-inferred storage (FIS), being a time integral of the total fluxes in and out of a region. This quantity highlights flux imbalances, seasonal cycles, interannual variability and trends very clearly and will also be used in developing the optimisations. For example for water we generate $\mathrm{FIS}_w$ using observations from the right hand side of Eq. (1).

If we have an initial storage (S0) we can generate a storage by integrating dS with respect to time. For water we take S0 to equal to GRACE storage from 1st January 2002 and use this to initialise the $\mathrm{FIS}_w$ time series;

$$\mathrm{FIS}_w = \int (\mathrm{dS})dt = \int (\mathrm{P} - \mathrm{E} - \mathrm{Q})^{(0)} dt. \tag{5}$$





If the initial fluxes were consistent with GRACE this should produce the GRACE storage time series, which prior to optimisation it does not. Also if there are any persistent imbalances in the fluxes this shows up as a strong trend in $\text{FIS}_w$.

A similar energy flux inferred storage can also be generated from the energy balance (Eq. 2), and we assume 0 storage initially;

$$\text{FIS}_e = \int (\text{NET})dt$$
$$\text{FIS}_e = \int (\text{DSR} + \text{DLR} - \text{USW} - \text{ULW} - \text{LE} - \text{SH})^{(0)}dt. \tag{6}$$

Again the $\text{FIS}_e$ will show up any seasonal cycle in surface warming, interannual variations, and long term imbalances very clearly.

### 3.2 Optimisation

Through an optimisation approach we produce monthly estimates of the water and energy budget components aiming to satisfy the following; 1.) minimise the distance from observed fluxes according to their relative uncertainties 2.) close the monthly water budget and long-term energy budget 3.) ensure the water and energy components are consistent 4.) ensure the total water storage implied from our optimised fluxes is with good agreement to total water storage from GRACE. The reasoning behind 4.) and methods to achieve this are described in detail section 3.2.1.

When combining observations from independent data products described in Sect. 2 we see an imbalance in the monthly water budget (Eq. 1). For the energy budget (Eq. 2) we generally do not have a monthly estimate of NET against which to assess imbalances, although we do have an expectation that the long term mean NET = 0, however consider for the moment that monthly NET is also constrained. If we write the monthly water and energy budget variables in a column vector $\boldsymbol{F_{obs}}$, where subscript 'obs' denotes observed values, then Eq. (1) and (2) can be expressed as a linear function of $\boldsymbol{F_{obs}}$. Let $\boldsymbol{A}$ and $\boldsymbol{B}$ represent the water and energy budgets respectively

$$\boldsymbol{AF_{obs}} = r_w \neq 0 \tag{7}$$
$$\boldsymbol{BF_{obs}} = r_e \neq 0 \tag{8}$$

$r_w$ represents the water budget residual and $r_e$ represents the energy budget residual, which we will also write together as residual vector $\boldsymbol{R}$ appearing later. The optimisation acts to adjust the observed fluxes to close the budgets by redistributing $r_w$ and $r_e$ to get $\boldsymbol{R} = 0$. We aim to find a new column vector $\boldsymbol{F}$ containing the optimised estimates we seek,

$$\boldsymbol{F} = \boldsymbol{F_{obs}} + \boldsymbol{a} \tag{9}$$

Where $\boldsymbol{a}$ is a vector of the same size as $\boldsymbol{F}$ containing adjustments, such that $\boldsymbol{Aa}$ = -$r_w$ and $\boldsymbol{Ba}$ = -$r_e$. In order to calculate $\boldsymbol{F}$ for month k, a cost function is setup as follows



$$J_k = \frac{1}{2}(\boldsymbol{F} - \boldsymbol{F_{obs}})\boldsymbol{S_{obs}^{-1}}(\boldsymbol{F} - \boldsymbol{F_{obs}})^T + \lambda\boldsymbol{AF} + \mu\boldsymbol{BF}. \tag{10}$$

Closure constraints are imposed via Lagrange multipliers ($\lambda$ and $\mu$). $\boldsymbol{S_{obs}}$ is a covariance matrix containing flux variances on
the diagonals. The off-diagonal elements would represent error covariance between input fluxes. In nearly all previous literature
260 (Sheffield et al., 2009; Abolafia-Rosenzweig et al., 2021; Hobeichi et al., 2020) the covariance matrix is assumed to be diagonal
as shown in (Eq. 11), although correlated errors may well be present due to the structural assumptions used for deriving Earth
Observation (EO) based surface fluxes. We will discuss the potential impact of such error covariances in Sect. 6.

$$\boldsymbol{S_{obs}} = \begin{bmatrix} \sigma_P^2 & 0 & \dots & 0 \\ 0 & \sigma_Q^2 & \dots & 0 \\ \vdots & \vdots & \ddots & \vdots \\ 0 & 0 & \dots & \sigma_{NET}^2 \end{bmatrix} \tag{11}$$

265 The cost function is minimised by setting the derivative with respect to each variable (F, $\lambda, \mu$) to zero. This results in the
following constraints

$$(\boldsymbol{F} - \boldsymbol{F_{obs}})\boldsymbol{S_{obs}^{-1}} + \lambda\boldsymbol{A} + \mu\boldsymbol{B} = 0 \tag{12}$$
$$\boldsymbol{AF} = 0 \tag{13}$$
$$\boldsymbol{BF} = 0. \tag{14}$$

270 These constraints are then used to calculate values for $\mu$ and $\lambda$, and solve for $\boldsymbol{F}$ via the least squares method.

The equations above allow to balance water and energy budgets each month and can be solved independently every month,
as has been done in previous literature (Pan et al., 2012; Abolafia-Rosenzweig et al., 2021; Hobeichi et al., 2020). However
the resulting solutions will not necessarily give sensible longer timescale water or energy budgets. The optimised values of dS
275 would not integrate to give a sensible trend or follow the observed variations of GRACE on longer timescales. Similarly there
is nothing to ensure that the integrated NET energy flux would remain realistically small, even if a monthly NET flux prior
were available to use in the optimisation.

### 3.2.1 Sequential method for water budgets

One approach to imposing longer timescale constraints on the solution would be to make a single optimisation over all months
280 together simply by extending the $\boldsymbol{F}$ vector, and including additional constraints on the sum of all the monthly water and energy
storage changes. This may work well for energy where the only long term constraint is on the NET energy flux, and is the
approach used by NEWS (L'Ecuyer et al., 2015), however it would still not allow to follow seasonal to interannual water
storage information present in the GRACE data.




Instead we opt for a sequential monthly approach. The monthly budget solutions are then not independent but take stock of previous optimisations as well as the observed GRACE storage change from the start of the period up to the present time. The optimisation acts to minimise the distance of the $\text{FIS}_w$ generated from optimised fluxes with GRACE storage change at the end of each month, according to GRACE uncertainties. This constraint requires a term in the cost function of the form;

$$(\text{FIS}_w^k - \text{S}^k)^2 \sigma_{dS}^{-2}, \tag{15}$$

which must be adapted in order to solve for dS. For an arbitrary month $k$ the optimised $\text{FIS}_w$ will be equal to the optimised $\text{FIS}_w$ for month $k-1$ plus the optimised dS for month $k$

$$\text{FIS}_w^k = \text{FIS}_w^{k-1} + dS^k. \tag{16}$$

By using Eq. (16) to rewrite $\text{FIS}_w^k$, we produce a term compatible with our cost function (Eq. 10) in order to impose the constraint described by Eq. (15) whilst solving for $dS^k$

$$(\text{dS}^k - (\text{S}^k - \text{FIS}_w^{k-1}))^2 \sigma_{dS}^{-2}. \tag{17}$$

Note that implementing this constraint only requires adapting the $\boldsymbol{F_{obs}}$ vector in the cost function. This requires $\text{FIS}_w^{k-1}$ to be known when solving for month $k$, which is only possible when solving sequentially. The optimisation is performed for all months between January 2002 and October 2013.

### 3.2.2 Sequential method for Energy budgets

Returning to Eq. (2) we note that generally we have no monthly constraints on either the surface energy storage or on the NET energy flux which could be used in a monthly optimisation. Although local measurements of ground heat flux are available from some flux tower sites, which have been used in previous energy budget studies (Hobeichi et al., 2020), these NET fluxes are very poorly observed, are associated with large uncertainties even locally, and are not available at basin scales. We have chosen not to use any independent NET prior and will comment on the consequent variability in surface energy storage results.

We do apply a minimal constraint on the prior monthly energy fluxes aimed only to give a small long-term NET energy storage change during optimisation. To do this we make use of the $\text{FIS}_e$. When combining the observed energy fluxes to obtain NET and averaging over the whole time period we see large imbalances, varying by basin, shown in Fig. 2. Therefore when integrating NET over time to infer the $\text{FIS}_e$, large trends are generally found, see Fig. 8. First we detrend the $\text{FIS}_e$, which is equivalent to removing the mean NET flux, to close the long-term energy budget, while preserving any interannual and seasonal variability. This detrended $\text{FIS}_e$ ($\text{FIS}_e^D$) is then used as a sequential monthly energy storage constraint in the same approach used to constrain the long term water storage changes to GRACE in the previous section, using a cost function term during month $k$ on $\text{NET}^k$;

$$(\text{NET}^k - (\text{FIS}_e^{Dk} - \text{FIS}_e^{k-1}))^2 \sigma_{NET}^{-2}. \tag{18}$$




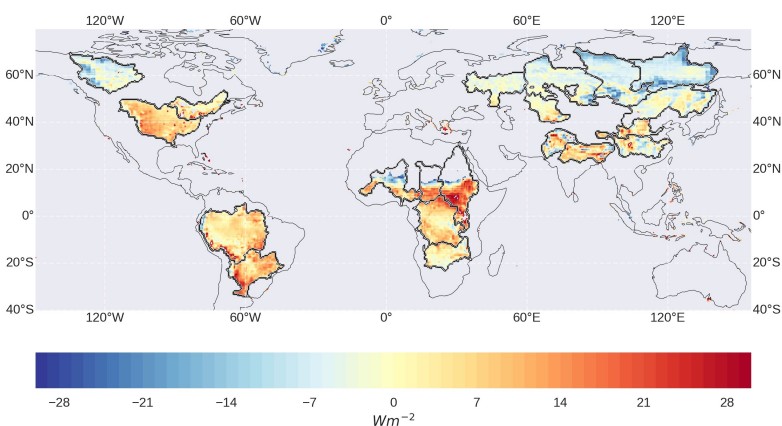

**Figure 2.** NET downward energy flux derived from CERES radiative fluxes and latent and sensible heat fluxes from FluxCOM, averaged over 2002-2013.

This $FIS_e^D$ based on the original fluxes, plays a similar role as the GRACE water storage change observations from the start to the current month, and ensures that the optimisation removes the NET energy trend over 2002-2013 without providing any further constraints on monthly to interannual variability for any of the component fluxes. The $\sigma_{NET}$ uncertainty we use here can be large and we chose a value equivalent to the combined component flux uncertainties expressed in Eq. (2). This has the advantage to ensure the optimised energy fluxes do not lead to any divergence in surface energy storage. However, it does assume there is no change in energy storage in this time period, which is not backed by any additional data, such as land surface temperature that might give more information about energy storage anomalies.

### 3.2.3 Temporal smoothing

Tight closure constraints imposed during the optimisation can result in high-frequency oscillations in the optimised flux solutions (Pellet et al., 2019), particularly for the water budget. Therefore we applied some temporal smoothing to the input observations to de-noise the time series, although this may also suppress some information. Pellet et al. (2019) use a similar (although slightly smoother), filter and concludes after comparison with other filters that it is a good compromise between these two affects of smoothing.

GRACE and energy storage is smoothed with weights $\frac{1}{8}, \frac{3}{8}, \frac{3}{8}, \frac{1}{8}$ which is equivalent to smoothing monthly changes, dS, NET, with the central weights $(\frac{1}{8}, \frac{1}{4}, -\frac{1}{4}, -\frac{1}{8})$ used by Eicker et al. (2015). P, Q, E and energy fluxes are smoothed using weights $\frac{1}{22}, \frac{1}{4}, \frac{9}{22}, \frac{1}{4}, \frac{1}{22}$. This choice of weights ensures that the amplitude of a sinusoidal signal would be damped in exactly the same way as is being applied to storage changes, so that the right and left sides of Eq. (1) and Eq. (2) would be treated the same, (Eicker et al., 2015).





At the time of this study, all data was available to us from January 2001 until December 2013. As this selected method of smoothing requires values from two preceding and two following months, our smoothed time series ends October 2013. Averages seen later in results include only complete years (January 2002-December 2012).

## 4 Results

### 4.1 Water Fluxes

Figure 3 shows both the input and optimised water fluxes over 3 large basins, the Amazon, Congo and Mississippi basins, on a monthly (right) and as a mean seasonal cycle (left). The adjustments made by the optimisation in order to balance the water budget are always small and usually within 1 standard deviation (SD) of initial uncertainties. To give an idea of the imbalance, monthly residuals are also shown, and the root mean square (RMS) of these residuals is around 40-45 % of the RMS of the GRACE storage changes and around 5-15 % of the RMS of precipitation, for each basin. Any interannual variability present in the observations is retained by the optimised fluxes.

Over the Amazon the seasonal cycle in precipitation largely converts directly into storage variations, with a smaller seasonal runoff signal lagging by around 3 months reflecting the large basin size and slow runoff. Evapotranspiration is almost constant through the year reflecting the constantly moist rain forest conditions, with the very small adjustments making E even more uniform. The residual in the water budget also shows a regular seasonal cycle, but anti-correlated to precipitation. The optimisation acts to increase P and decrease Q from November-March when precipitation peaks, while the adjustment is mainly an increase in Q from June-August, which has the effect of prolonging the runoff peak. where the adjustment occasionally exceeds 1 SD. For these months there are lower uncertainties on P and E, hence most of the residual has been distributed to Q.

The Congo's precipitation is characterised by biannual peaks as the ITCZ migrates across the Equator. The primary maxima occurs towards the end of the year and the secondary maxima occurs in May. The bimodal peaks are also seen in the Q and E fluxes. There is also considerable interannual variability in the Precipitation. The optimisation adjustments are not easily summarised as a regular seasonal pattern.

Unlike the other two basins, over the Mississippi the storage changes are almost out of phase with the precipitation. This reflects the dominance of storage in snow as a key controlling mechanism. The maximum runoff then occurs before the peak precipitation, indicative of snowmelt followed by early summer rains. Much of the seasonal Precipitation peak is balanced by evapotranspiration which exceeds precipitation in July and also coincides with the largest reductions in storage. There are some very low precipitation years such as 2006 and 2012. Due to the larger role of evapotranspiration in this basin, the optimisation also shows a consistent E increase from July-December each year. We will look at this in more detail when considering the coupling to the energy budget.



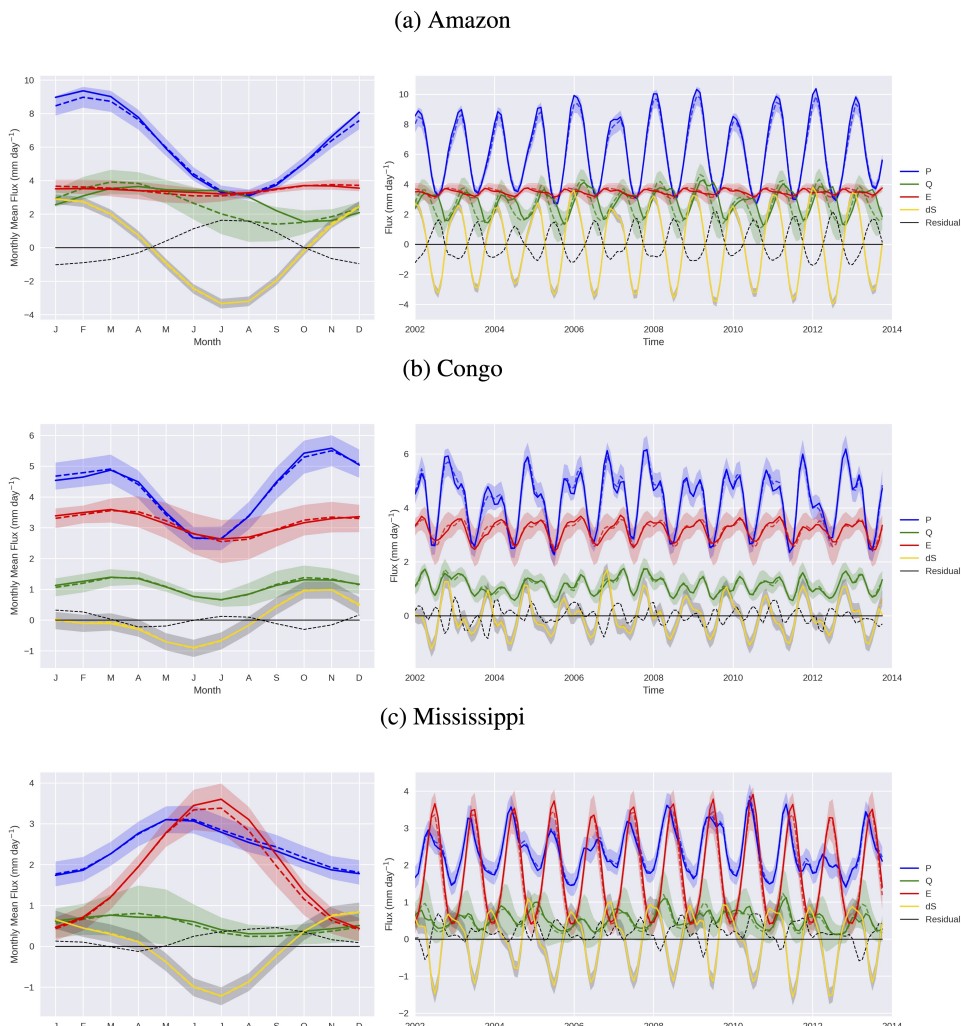

**Figure 3.** Water budget fluxes. Observed values are shown by the dashed lines and optimised values from our study are shown by the solid lines. The shaded regions show uncertainty of the observed values. Observed fluxes and derived products shown here and later plots are all temporally smoothed as described in Sect. 3.2.3. Mean seasonal cycles are shown left.

## 4.2 Total water storage

Greater insight into the fitting process and the changes involved are illustrated best by plotting the time history of surface water storage for a number of basins (Fig. 4). The left hand plots show the GRACE storage "target" variability in orange and the flux-inferred storage (FIS$_w$) from raw observations, dashed purple. For some basins these already match reasonably well e.g.

365 Congo, but for others the raw observations show a large trend and the FIS$_w$ disappears off plot. Further details can be seen when the adjusted FIS$_w$ is made consistent with the GRACE changes over the whole period, solid purple. Over the Amazon it





is clear that the seasonal storage cycle represented by the fluxes is weaker than indicated by GRACE. Over the Amur the flux derived seasonal cycle in contrast is large compared to GRACE storage. Several of the basins also show significantly different interannual variability. The plots on the right show that the storage based on the optimised fluxes now sit very clearly on top of the GRACE storage data and fit the seasonal amplitudes and interannual variability in all basins very well. The differences compared to GRACE data are also shown and these are always smaller than the 1 SD uncertainties applied to GRACE during optimisation.

### 4.3 Storage comparison with other products

We find that the total water storage is a useful metric to compare against other optimisation products that have been made available in the literature. We take the monthly water storage change, dS, products from three different recent budget closure studies and calculate the total water storage that these imply. The CLASS product (Hobeichi et al., 2020) provides a complete set of balanced coupled water and energy budget components on a global grid for the period 2003-2009, and we will later also compare with the energy budget from this solution. The CDR (Zhang et al., 2018), provides grid-point estimates of monthly closed water budget components from 1984-2010, which includes a GRACE constraint over the later period. The RESEEN product (Abolafia-Rosenzweig et al., 2021) used three different closure methods, and for this comparison we take the ensemble mean from the combined "proportional distribution" (PR) method, which was described to give the best results.

We calculate the total water storage for the period 2003-2009, based on the overlap of these three products, and plot these against GRACE and our optimised solution in Figure 5. Each storage has been initialised with GRACE for January 2003.

It is clear that while all of the other products have quite similar storage variability over each year, they all show some degree of divergence from GRACE storage over longer timescales. In the corrected RESEEN dataset, mean product corrections show that the closure constraints act to increase observed dS around 3 mm per month on average. This adjustment may result in an upwards trend in storage not observed by GRACE in some regions. For example, the RESEEN product shows an upward storage trend over the Mississippi and Huang He basins, although over the Congo the storage fits GRACE quite well. The CLASS product shows an upward trend in water storage in all 3 basins and therefore shows storage differences at the end, reaching 14 cm equivalent over the Congo. The CDR product probably does best in fitting the 7 year trends for all 3 basins as a long period water balancing correction is applied, (Zhang et al., 2018), however it shows anomalously weak seasonal variability in some years over the Mississippi and the Huang He, and also misses some of the interannual variability over the Congo. The constraint we apply of fitting GRACE storage on all timescales can again be clearly seen.

Although these other studies may have used different choices of GRACE product as constraints, after comparison we find that all GRACE products are very similar and the differences shown here are coming from the optimisation approach.

For the Mississippi the CDR shows good agreement with GRACE, although it shows reduced seasonal cycles after 2008. CLASS and RESEEN show a slight positive bias compared to GRACE after 2005 but generally agree well with the size of the seasonal cycle. Over the Yellow River GRACE shows a large peak in water storage followed by a decrease in storage, amounting to around 10 cm of water loss by 2009 since the peak in 2003. All products capture the initial peak to some extent but fail to detect the downwards trend. The interannual variability of the seasonal cycle observed by GRACE is represented well





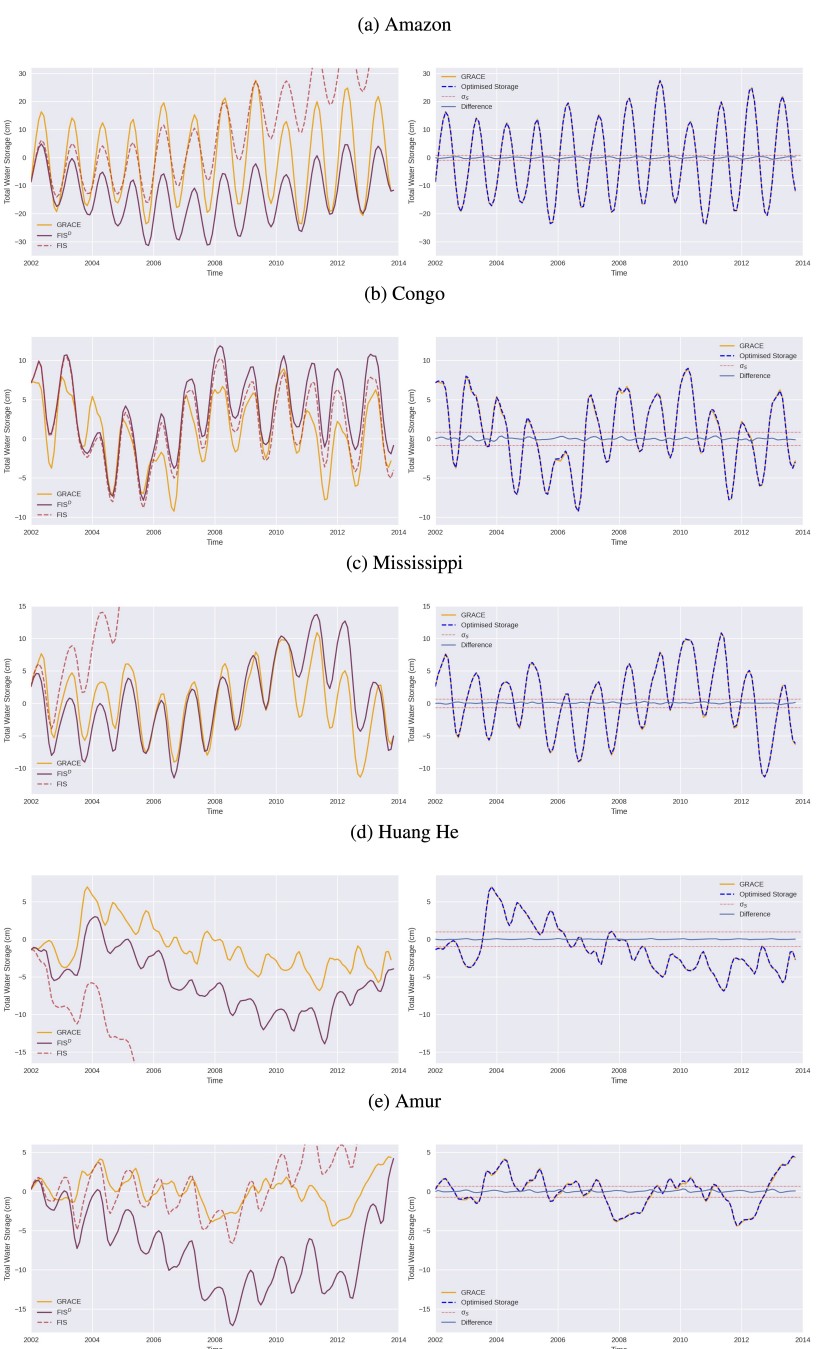

**Figure 4.** GRACE storage compared with Flux inferred storage. Left: The unadjusted $\text{FIS}_w$ is dashed red line and the detrended $\text{FIS}_w^D$ is the solid purple line, generated directly from observed fluxes. Right: The "Optimised storage" based on the the new fluxes from our study is seen to closely follow the GRACE storage which is shown in orange on the same axes in both plots.





**Figure 5.** Water storage product comparisons from 2003-2009 inclusive. The "Our Optimised Storage" line is the result from our study.





by the RESEEN product. Such as the reduced seasonal cycle between 2006 and 2007, and the larger seasonal cycle in 2003. The timing of the seasonal cycle also shows good agreement. RESEEN is able to detect the decline in storage 2004-2007 well, however does not capture the decline 2008-2009. The CLASS product does not show good agreement with the overall trend in water storage. By 2009 there is over 6 cm difference between GRACE and the CLASS storage since the downwards trend of

GRACE was not captured nor accounted for. The CDR consistently shows reduced seasonal cycles. Some of the interannual variability is captured but it does not agree with the long-term trend. Since the CDR imposes a constraint which ensures dS = 0 over 1984-2010, it means that the storage in 2010 must be the same as 1984, hence limiting the ability to detect trends.

### 4.4 Energy Fluxes

The optimised energy components in Fig. 6 show only small differences from the observed fluxes used as input, although the

long term NET energy budget is now closed through the constraint coming from $\text{FIS}_e^D$. To see the adjustments more clearly Fig. 7 shows the seasonal mean adjustments to the NET downward flux and the component fluxes in 3 of the basins. The energy closure clearly requires a reduction of the NET downward energy flux in each of these basins in all months. Most flux components contribute fairly uniformly to the NET change except for some variations responding to LE adjustments imposed through the water cycle. In all 3 basins the adjustments to LE modulate NET changes through the year. In both the Amazon and

the Mississippi the adjustments through the water cycle are having a small dampening effect on the seasonal cycle in NET flux, as can be seen in Fig. 6. If any independent data on monthly NET flux or storage were to be available this would potentially change these monthly flux adjustments considerably, as we will discuss later.

### 4.5 Total Energy Storage

As for the water cycle we also find that total surface energy storage provides a useful metric to see the impact of the optimi-

sation. We also compare energy storage from our optimisation with results from CLASS (Hobeichi et al., 2020), which is the only other study to provide coupled regional water and energy budgets at a monthly timescale.

There is a key difference in the constraints imposed on the energy budget components in this study and CLASS. CLASS enforces monthly closure and uses measurements of ground heat flux (equivalent to NET) as input for their optimisation, whereas in our optimisation we do not use ground heat flux observations as input.

Figure 8 shows, for several basins, the interannual NET ground heat flux (left) and the implied surface energy storage, and the $\text{FIS}_e$ (right). Shown for the period January 2003 to December 2009, which is the time frame used in the CLASS study.

The energy storage plots include the $\text{FIS}_e$ before the fluxes were detrended as well as our optimised solution and the CLASS solution. In all basins our initial NET energy fluxes are unbalanced and show a strong storage trend. The CLASS solutions also show smaller, but still potentially unrealistic, energy storage trends in all basins apart from the Mississippi, because CLASS

does not account for energy imbalances on timescales longer than 1 month.

Both CLASS and our optimised fluxes show clear seasonal cycles of warming and cooling in the mid latitude basins of Mississippi, Amur and Huang He rivers. There is much more variability in our NET fluxes while the CLASS fluxes are similar every year, presumably reflecting the dampening effect of a ground heat flux constraint in CLASS. Our optimised solutions

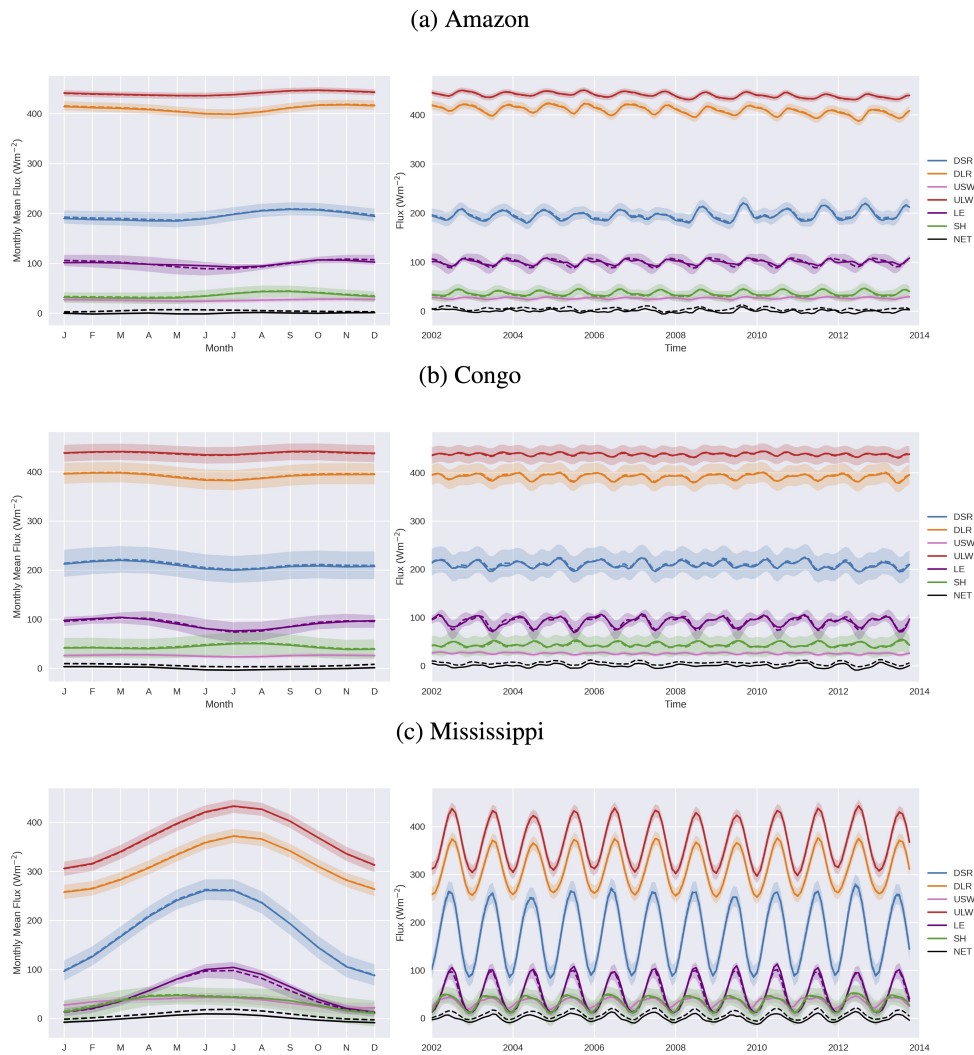

**Figure 6.** Energy budget fluxes. Observed values are shown by the dashed lines and optimised values are shown by the solid lines. The shaded regions show uncertainty of the observed values. Mean seasonal cycles are shown left.

show more interannual flux and storage variability than CLASS, although this would not amount to any trend if the timeseries
were extended to 2013, over which time we have used the detrending energy constraint. This interannual variability is inherent
in the initial energy fluxes, in particular from the radiation components, and is not in general being introduced through water
cycle coupling. We will return to discuss this seasonal and interannual energy storage variability later.



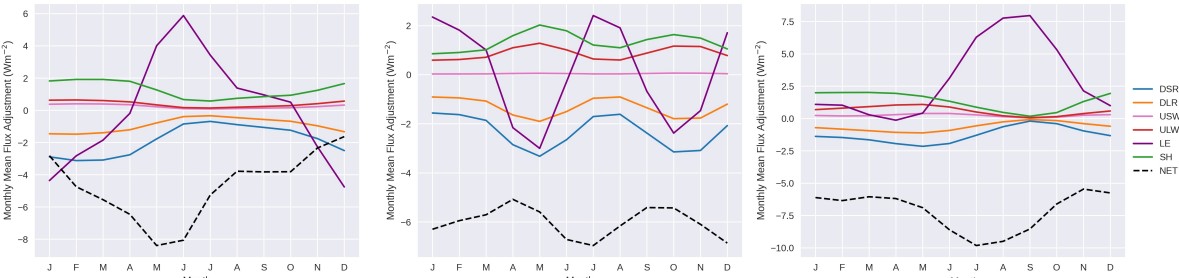

**Figure 7.** Optimisation adjustments for the energy components. Optimised values - observed values. Amazon (left), Congo (centre), Mississippi (right).

## 5    Optimisation Uncertainties

### 5.1    Goodness of fit

The consistency of the optimisations with the uncertainties provided are expressed by the $\chi^2$ measure, which are summarised in Table 2. Generally $\chi^2$ should be smaller than the number of independent variables being constrained. Each month the 4 water variables expressed in Eq. (1) are required to balance. Although the energy budget Eq. (2) is coupled and must also balance in the long term, without an independent constraint on the NET monthly flux or energy storage change, these variables contribute very little to $\chi^2$, and the Total $\chi^2$ are only marginally larger than the Water only values. It can be seen that the average values

are always much less than 4, and remain smaller for the maximum value in any individual month. We conclude that all the sequential optimisations are providing solutions consistent well within the given uncertainties.

| Basin | Total $\chi^2$ mean | Total $\chi^2$ max | Water $\chi^2$ mean | Water $\chi^2$ max |
|---|---|---|---|---|
| Amazon | 0.98 | 3.09 | 0.92 | 2.94 |
| Congo | 0.21 | 1.18 | 0.18 | 1.10 |
| Mississippi | 0.28 | 1.74 | 0.26 | 1.73 |
| Amur | 0.21 | 0.67 | 0.16 | 0.67 |
| Yellow River | 0.11 | 0.39 | 0.08 | 0.36 |

**Table 2.** The $\chi^2$ values for the optimisation include the Total values, including all the water and energy flux terms, and with only the 4 Water adjustment terms which are more strongly constrained at monthly timescales. The time mean $\chi^2$ over all months, and the largest value for any individual month, are also shown.

### 5.2    Uncertainty estimates

By using multiple datasets to constrain each other through budget closure, the uncertainty of the optimised estimates will be less than the uncertainty of the raw observations. The uncertainties on the new estimates are calculated using the same methods





(a) Amazon

(b) Congo

(c) Mississippi

(d) Amur

(e) Yellow River

**Figure 8.** NET fluxes (left) and total energy storage $FIS_e$ anomalies (right) 2003-2009. The CLASS solutions are compared with our optimised solutions, with the $FIS_e$ also shown prior to optimisation, right.



as the NEWS study, given by $\mathbf{S_F} = (\mathbf{K^T S_R^{-1} K + S_{Fobs}^{-1}})^{-1}$. Where $\mathbf{K}$ is the Jacobian of $\mathbf{R}$ with respect to $\mathbf{F}$, and $\mathbf{S_R}$ is the uncertainty on residual constraint $\mathbf{R} = 0$. Since we use a strong constraint to impose budget closure the uncertainty $\mathbf{S_R}$ is small. The uncertainties in the water budget terms before and after optimisation are now shown in Table 3.

| Basin | $\sigma_{P_{obs}}$ | $\sigma_P$ | $\sigma_{Q_{obs}}$ | $\sigma_Q$ | $\sigma_{E_{obs}}$ | $\sigma_E$ | $\sigma_{dS_{obs}}$ | $\sigma_{dS}$ |
|---|---|---|---|---|---|---|---|---|
| Amazon | 0.47 | 0.40 | 0.77 | 0.46 | 0.37 | 0.34 | 0.29 | 0.28 |
| Congo | 0.45 | 0.37 | 0.29 | 0.26 | 0.51 | 0.38 | 0.28 | 0.26 |
| Mississippi | 0.31 | 0.27 | 0.41 | 0.31 | 0.39 | 0.30 | 0.22 | 0.21 |
| Amur | 0.36 | 0.33 | 0.42 | 0.36 | 0.66 | 0.43 | 0.23 | 0.22 |
| Yellow River | 0.65 | 0.60 | 0.79 | 0.70 | 1.35 | 0.84 | 0.32 | 0.31 |

**Table 3.** Average monthly water budget component uncertainties in mm /day before (obs) and after the optimisation.

It can be seen that uncertainties reduce typically by 10 % for precipitation but by substantially larger amounts for runoff and evapotranspiration where initial errors are larger. The uncertainties in storage change are only marginally affected. Of course formally, post optimisation uncertainties are correlated reflecting a closed water budget with no residual.

## 6 Discussion

The initial imbalances in the water and energy budgets vary by basin. We found water budget residuals ranging between 1.5 % and 35 % of precipitation, which is comparable to Abolafia-Rosenzweig et al. (2021) who found that residual errors varied between 0.7 % and 30 % of precipitation. These initial imbalances are dependent on the quality of the input data, which differ according to the geophysical characteristics of the basins.

The Amazon has the largest absolute water budget residual prior to optimisation, averaging $0.86 \ \mathrm{mmday^{-1}}$. Sahoo et al. (2011) also identified the Amazon as having the largest non-closure error and suggested this could be as a result of the sparseness of in situ precipitation measurements over the basin which are required for the calibration of satellite estimates. Of the five basins in Fig. 8, the Amur showed the greatest water budget imbalance as a percentage of precipitation, with a monthly average of 28 % and a maximum of 77 %. The Amur is situated between latitudes 45° and 55°N, and so the large imbalance could also be due to a lack of observational data. In particular precipitation tends to be less well observed at higher latitudes since key satellites such as the TRMM focus only on the tropics and sub tropics. The optimisation makes a 7 % adjustment to P for the Amur, the largest normalised adjustment of these basins, reflecting the large specified uncertainty. The Congo showed the lowest initial imbalance of these basins, normalised with respect to P, with an monthly average of 6 % and a maximum of 19 %.

In previous budget closure studies longer timescale constraints on the water budget have often been overlooked. This can result in substantial misfits against the GRACE storage timeseries, particularly for regions which show significant trends and interannual variability. The sequential optimisation approach used here is beneficial as it enables dS to be constrained



by GRACE on all timescales and guarantees that the total water storage implied from the optimised fluxes will track the
interannual variability of GRACE as well as avoiding any unrealistic trends.

However the sequential solution method does not permit flux adjustments across more than one month at a time. It is possible
to make a whole period adjustment, closing the water budget every month while imposing a small or zero trend in the water
storage from beginning to end of the timeseries. This would allow adjustments across months to fit longer term changes, and
has been used in solving the seasonal cycle in the NEWS solution of Rodell et al. (2015) for example. However this still does
not guarantee a fit to the interannual variability information present in the GRACE timeseries. We made some comparisons
optimising for all months together (results not shown) and this worked well for some basins (e.g. Mississippi) and is then very
similar to the sequential solution results, but it works much less well for other basins (e.g. Congo) when interannual variations
are seen in the GRACE timeseries.

Our method also allows the optimised energy fluxes to be in good agreement with the initial energy flux observations
whilst also balancing the monthly water budget and removing long term energy trends. However, the lack of a monthly NET
energy constraint means that the energy budget is only very weakly constrained on short timescales. Further observational
information such as land surface temperatures, along with a heat capacity, could be used to constrain the energy storage on
these timescales. Liu et al. (2017) propose NET heat flux estimates from ECMWF reanalyses based on surface temperatures
and some land surface modelling. Alternatively some estimate of monthly NET ground heat flux upscaled from flux tower
measurements could be imposed, as in Hobeichi et al. (2020). While we made some comparisons here, we have preferred to
leave the monthly energy budget fairly unconstrained as other monthly NET energy flux products have not been adequately
validated for use as independent data. This also allows surface variability inferred from other flux products to be clearly seen,
such as in Fig. 9.

We have noted above that the differences between our NET energy fluxes, and those report by the CLASS product, is likely
due to the CLASS product including a ground heat flux, G, product in its formulation. G is the least well observed of all the
energy balance fluxes, with typical measurements covering only very small areas. Consequently, large scale gridded products
tend to contain high levels of uncertainty due to errors of representivity in the underlying data, and hence rely on modelling
assumptions that can have a strong influence on the resulting flux estimates. Consequently we chose not to include a data set
of G in our budget modelling.

Results shown have all assumed initial errors are uncorrelated. However, due to the procedures required to derive some of
the flux products, it is likely that not all fluxes are fully independent. For example, the GRUN product partly predicts runoff
based on antecedent precipitation conditions and so any error in P may also be present in Q. Specifying an error covariance
(off-diagonal elements in Eq. 11) impacts how the fluxes are adjusted during the optimisation, and also reduces the effective
number of independent variables. We performed some sensitivity tests applying error covariances through the covariance matrix
($\mathbf{S_{obs}}$). To give an example, consider the P and Q errors to be correlated. Adjustments needed to close the water budget Eq. (1)
would normally require P and Q to be adjusted in opposite directions e.g. a smaller P and larger Q would both reduce a positive
budget residual. However correlated P and Q errors would tend to inhibit an opposite adjustment of this sort. In consequence,





imposing correlated P, Q errors will lead to smaller adjustments in both P and Q and require the other budget terms, E and dS, to have larger adjustments in order to close the budget. This is demonstrated in the results.

The same arguments apply for correlated errors in the energy fluxes. If upward and downward radiation flux errors are positively correlated this will reduce the degrees of freedom, reduce the adjustments in those fluxes and increase budget balancing adjustments to other fluxes. Correlating Sensible and Latent flux errors (both upward fluxes) however, as implied by eddy covariance studies, e.g. Twine et al. (2000), will increase their adjustment contributions by reducing their joint cost function impacts. As all adjustments we found were well within error bounds in these regional solutions we did not find any incon-

sistencies when imposing realistically correlated initial errors. However it is worth commenting that if flux component error correlations are present they may be quite pervasive and would then imply larger or smaller adjustments to large scale energy fluxes; changing, for example, the relative adjustments to radiation compared with turbulent fluxes in global inverse budgets such as described in L'Ecuyer et al. (2015).

**7   Conclusion**

This study has introduced a sequential optimisation approach which is used to produce coupled estimates for the components of the terrestrial water and energy budgets based on observations. The focus has been on several large river basins over the period 2002 to 2013. The novelty of this optimisation is that it acts to close the monthly water budget while at the same time matching the water budget on longer timescales. This then achieves a good fit with the GRACE surface water storage timeseries in each

basin when the optimised fluxes are integrated, which a number of previous products do not. The coupled energy budget is also solved sequentially while still guaranteeing a long term energy balance. This is achieved using a detrended monthly energy storage target $\text{FIS}_e^D$ based on the original fluxes, as a weak constraint. All the flux adjustments made during the optimisation are small and within uncertainty estimates, and interannual variability from observations is retained. The optimisation also has the benefit of reducing formal uncertainties on the individual flux components.

We show the flux-inferred storage (FIS) for both water and energy gives a sensitive measure of the imbalances, seasonal cycles, and interannual variability amplitudes implied by the fluxes that can then also be compared with GRACE and with several other products from the literature. For several basins, the input water fluxes show weaker of stronger seasonal amplitudes than is suggested by GRACE, which are then corrected during the optimisations.

The current study has focused on methods for budget balancing adjustments. We have not used a selection of different input data products to test the relative imbalance from different choices. Also budgets are only balanced on a selection of larger land hydrological basins. Figures relevant to more basins are included in the Appendix. We have not produced a gridded product of optimised fields although this could be done at some resolution consistent with the resolution of the input products, in particular the GRACE data.

Although the energy budget is coupled the current solutions are only constrained on long timescales. Flux components adjust to a long term surface energy balance, accounting for any mean changes in latent losses imposed through the water cycle, but otherwise monthly energy components are relatively unconstrained. Further work could seek to constrain the surface energy fluxes on shorter timescales by introducing additional energy storage data e.g. using land surface temperatures either from EO or reanalysis products (Liu et al., 2017).

Another direction of work could seek to include a coupled water and energy budget for the atmosphere. This could be build into a global solution as in NEWS (Rodell et al., 2015; L'Ecuyer et al., 2015) or else would need to include regional boundary transport estimates in the atmosphere for both energy and water (Mayer et al., 2022).

    Overall, the constraints imposed as part of this study and the direction of future work are aimed at improving the accuracy of water and energy cycle components, which can ultimately help us gain a better understanding of climate processes and improve

the skill of climate models in predicting future change.

*Code and data availability.* The observational data used as input is available from several different sources. GRUNv1 runoff data can be obtained from https://www.bafg.de/GRDC/EN/04_spcldtbss/43_GRfN/refDataset_node.html, GPCPv2.3 precipitation data can be obtained from https://doi.org/10.7289/V56971M6, turbulent heat flux data from FluxCOM can be obtained from ftp://ftp.bgc-jena.mpg.de/pub/outgoing/FluxCom/EnergyFluxes/ , and CERES radiative flux data are available at https://ceres.larc.nasa.gov/data/. The CLASS product used

as comparison in this study is available from https://doi.org/10.25914/5c872258dc183, and the RESEEN and basin scale CDR product are archived on Mendeley Data and available at http://dx.doi.org/10.17632/r24rdxt73j.3 (Abolafia-Rosenzweig and Livneh, 2020).

    The optimisation code developed in this study is available at https://zenodo.org/record/7248284 (Petch, 2022)

*Author contributions.* This work was carried out by SP initially as part of her MSc project and subsequently as a contribution to her PhD. All the analysis and figures were generated by her as well as a first draft of all the paper text. KH, TQ and RK acted as PhD supervisors and

BD helped with data acquisition and analysis, and each contributed edits to the paper text.

*Competing interests.* The authors declare that they have no conflict of interest.

*Acknowledgements.* This work was supported by the NERC DTP SCENARIO program for Samantha Petch PhD. Haines, Quaife and Dong would also like to acknowledge support of the NERC NCEO International Programme (NE/X006328/1) for their contributions to this work.

## Appendix A

The following figures illustrate the results for a wider selection of basins across the globe, with varying initial imbalances.





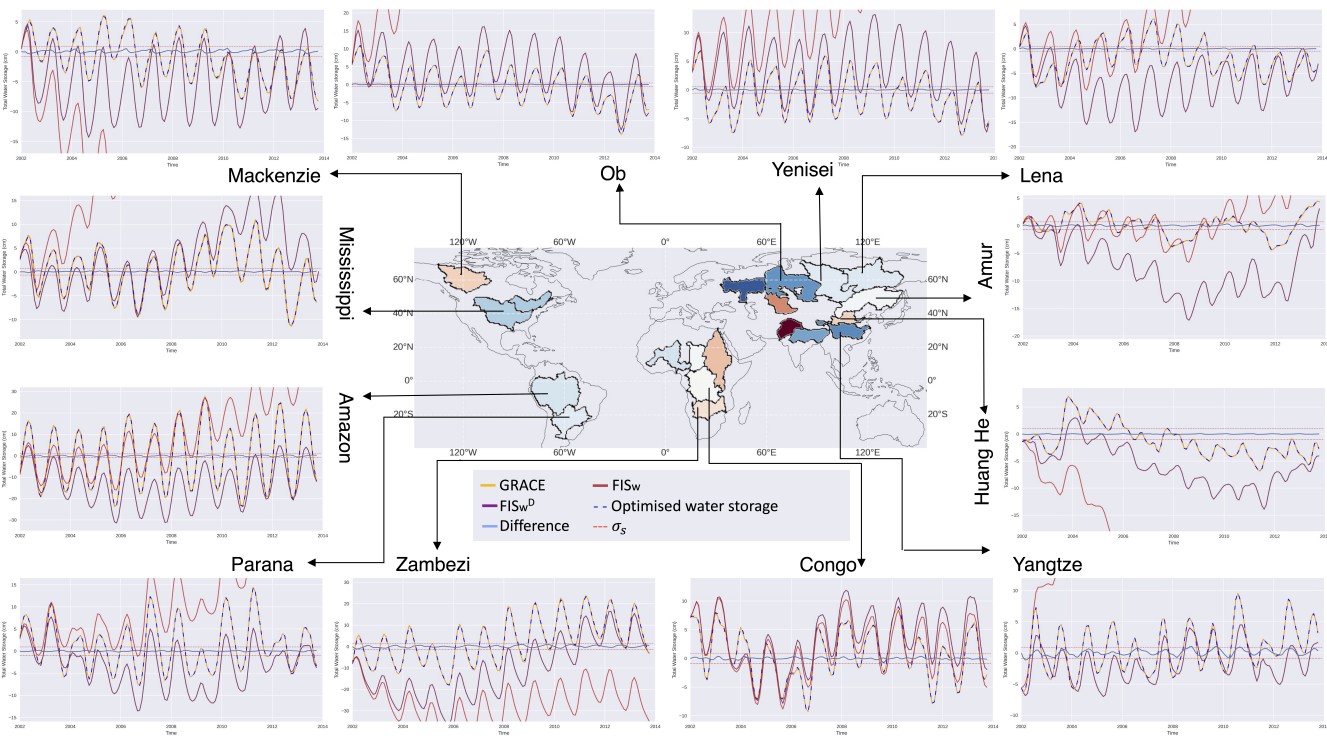

**Figure A1.** Monthly water storage variability in a selection of basins Jan 2002-Oct 2013. GRACE data is shown (orange) with the optimised water storage solution (blue dashed) overlaying it closely. The optimised differences to GRACE are also shown along with the GRACE uncertainties, $\sigma_S$, varying near zero. The figure also shows two versions of Flux Inferred Storage (FIS$_w$) from the input data. The original FIS$_w$ (red) is the storage implied by integrating the input (P-E-Q) in time. This often diverges rapidly demonstrating strong initial imbalances (the basins are coloured according to this initial imbalance). The Detrended FIS$_w^D$ (mauve) simply detrends those original fluxes to fit the mean GRACE storage trend. This reveals interesting details of the initial mismatch between water fluxes and GRACE, for example showing an underestimate in seasonal FIS$_w^D$ in the Amazon and a strong overestimate in seasonal storage in the high latitude rivers. Many basins also show mismatches in interannual variations compared to GRACE, which are all removed in the optimisation process.





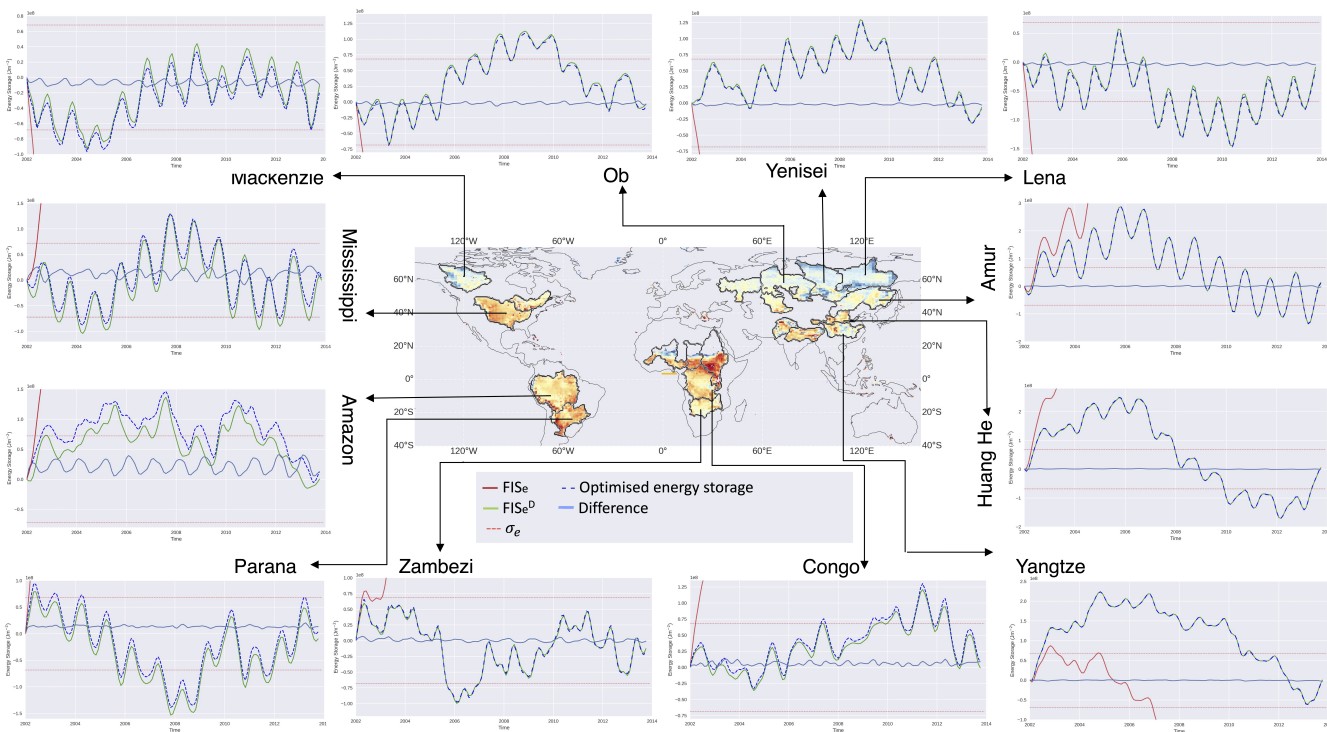

**Figure A2.** Monthly surface energy storage variability in a selection of basins Jan 2002-Oct 2013. The original Flux inferred storage $FIS_e$ (in red) is the storage implied by integrating the input (NET) in time. This always diverges rapidly demonstrating strong initial energy imbalances (the basin map is coloured according to these initial imbalances). The Detrended $FIS_e^D$ (green line) simply detrends the original NET flux to give a zero energy storage trend. This retains details of both seasonal and interannual surface energy variability. The implied interannual variability in surface storage is large compared to seasonal variations and can mostly be traced to variations in surface radiation fields. Such interannual variability may be unrealistic but without reliable observations of Ground NET heat flux or a measure of surface storage e.g. from Land surface temperatures, we have chosen to retain it during optimisation (see text for details). The optimized energy storage is also shown (blue dashed), and this broadly tracks the $FIS_e^D$ which is used as a monthly constraint. Any divergence is confined to the first few months so that the energy budget is closely balanced throughout the period. The difference in the optimized storage is also shown, along with the large uncertainty limits used, $\sigma_e$ (see text for details).



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
