# Peer review of "Water and Energy budgets over hydrological basins on short and long timescales."

_EGUsphere, 2022_

## Referee Comment (RC2)

**Review of Petch et al, EGUsphere, 2023**

Reviewer: Ruud van der Ent, Delft University of Technology

**General comments**

Petch and co-authors study the water and energy balance of 20 large river basins. Given my own expertise, I will mainly comment on the water balance part. Using GRACE-derived terrestrial water storage changes and globally available observation-based product of precipitation, evaporation and runoff they first study the imbalance between these products.

Then, they optimize their storage time series to nearly match that of GRACE and doing so update the fluxes as presented in Figure 3. Then, I failed to understand what the usefulness of the "Our Optimised Storage" and associated figures 4 and 5 are. It seems quite circular to me that if you force it to match GRACE it matches GRACE better than other products. From a hydrologist perspective I would rather see the optimized *P, Q* and *E* compared to other products. Perhaps even an independent better regional precipitation or river discharge dataset for specific basins, to see whether the optimized fluxes match that better and which would then clearly demonstrate the strength of their method. I am not sure if this is going to require major changes to the paper, or just a clearer explanation of the objectives and results.

A second major, but not difficult to solve issue, is that I find the authors somewhat sloppy regarding equations and symbology. Unfortunately, this set of guidelines has disappeared recently from the HESS manuscript preparations guidelines online: https://iahs.info/Publications-News/Other-publications/Guidelines-for-the-use-of-units-symbols-and-equations-in-hydrology.do, but I personally still appreciate it if we all try to follow this as much as possible. Of serious concern are Eq. (1) and Eq. (5), which should obviously read $dS/dt$ instead of $dS$ as the fluxes are per unit of time. It would also be helpful if the equations would contain the dimensions, thus, e.g. length^3 per time [$L^3 T^{-1}$] for Eq. (1) so this becomes obvious. Even expressed per unit area [$L\ T^{-1}$] would also be fine of course as long it clearly remains a flux and not a stock. Moreover, please use single italicized symbols, so something like $S_{fi}$ instead of FIS, which makes it directly clear we are talking about a storage. I know many other papers invent funny acronyms as well, maybe it is even the rule rather than the exception, but in my opinion, it is simply not pleasant for any reader.

A third major question is why the authors chose the data they chose and whether it matters for the main point they are trying to make. For precipitation and evaporation, many more observation-based products exist, so did they select the 'best' according to some previous studies or did they just select 'good' data and does it not matter a lot whether it is really the 'best'. I hope the authors can explain. Moreover, the runoff data is even dependent on precipitation and evaporation from GSWP3, which is a bit of a vague product in terms of how it was constructed and I think it may even rely partly on GPCP and FLUXCOM, making the estimates of *P, E* and *Q* not completely independent. Moreover, I fail to see why spatially varying runoff is necessary at all, as on the basin scale, the actual river discharge measurements at the river mouth would suffice for which, for example, the GSIM archive (Do et al., 2018) could have also been used.

I hope to authors to give a quick response and perhaps we can even already settle some issues in the open discussion phase.

**Specific comments**

L1-2: "improving climate and earth system models"

I would say 'validating' or 'assessing the capability of' which is to be done first before anything can be improved.

L6: "the corresponding turbulent heat fluxes ranges between ± 10 W m$^{-2}$"

I suppose something should range between value x and value y, thus this sentence misses something.

L8: "This exposes mismatches in seasonal water storage"

Mismatches between what and what exactly?

L12: "The optimization also reduces formal uncertainties on individual flux components"

Sounds great, but I failed to clearly identify this result in the paper itself.

L14: "The FIS metrics"

What are 'the FIS metrics'?

L23: "Water is a conservative quantity"

Technically speaking this statement is incorrect. Water is used by plants for photosynthesis and released by decomposition or fire. Probably it is an order of magnitude lower than the errors made in the products of *P, E* and *Q*, but not entirely negligible.

Table 1 "present" and general period statements

It is rather irrelevant whether e.g. GRUN is available until 'present' or that it starts in 1902, what matters is which years you used for the analysis.

L91 "evaporation"

I strongly support the use of evaporation over the ambiguous term evapotranspiration, see Miralles et al. (2020) for the arguments why that is, so perhaps you could simply use evaporation also elsewhere in the manuscript.

Equation 5

The integral is between what and what? What does the to the power 0 between brackets mean? Is this equation supposed to present a time series? Then it would be clearer if $S_{fi,w}(t)$ was explicitly written.

**Technical corrections**
L93: "Earths"

Earth's

L101: "Land"

land

**References**

Do, H. X., Gudmundsson, L., Leonard, M., and Westra, S.: The Global Streamflow Indices and Metadata Archive (GSIM) – Part 1: The production of a daily streamflow archive and metadata, Earth Syst. Sci. Data, 10, 765–785, https://doi.org/10.5194/essd-10-765-2018, 2018.

Miralles, D. G., Brutsaert, W., Dolman, A. J., and Gash, J. H.: On the Use of the Term "Evapotranspiration," 56, https://doi.org/10.1029/2020wr028055, 2020.

---

## Author Comment (AC1)

Thanks for giving a detailed review of our manuscript, we really appreciate all your comments. We would also like to thank you for encouraging a quick discussion and will use the opportunity to respond to your general comments. We will address the other comments in our response to all of the reviews later on.

**Then, I failed to understand what the usefulness of the "Our Optimised Storage" and associated figures 4 and 5 are. It seems quite circular to me that if you force it to match GRACE it matches GRACE better than other products. From a hydrologist perspective I would rather see the optimized P, Q and E compared to other products. Perhaps even an independent better regional precipitation or river discharge dataset for specific basins, to see whether the optimized fluxes match that better and which would then clearly demonstrate the strength of their method. I am not sure if this is going to require major changes to the paper, or just a clearer explanation of the objectives and results.**

'Our optimised storage' is the total water storage produced from integrating our optimised P, Q and E fluxes. The usefulness of this quantity is that it is able to capture information that cannot easily be seen when looking at individual monthly flux components. Small imbalances in monthly fluxes which would be hard to validate with any single month of data, can still imply unrealistic long-term changes in a basins water storage e.g. associated with rising/lowering water tables. GRACE data does track these longer timescales. For this reason, a set of fluxes consistent with GRACE is more useful for evaluating modelling products for example with which we may wish to attribute both short- and long-term water budget variations. Previous studies which have sought to develop observational flux products consistent with GRACE have failed to match low frequency variations which are important to understand through hydrological modelling.

And so, 'our optimised storage' is used to demonstrate how we are able to achieve long-term balance with only small adjustments to the monthly fluxes. Previous optimisation studies have only use GRACE on a monthly timescale. By using GRACE storage (S) rather than the monthly storage change (dS/dt) derived from GRACE, it enables us to use more information to provide stronger constraints than these other studies.

I have plotted the figure below to help explain this further. The figure compares two optimisation outputs for the Mississippi basin, both of which use GRACE as input. In blue, we show our sequential optimisation approach described in the study, along with an optimisation that enforces monthly water balance but with no long-term constraint in green. The left figure shows monthly flux imbalances (P – E – Q) compared to the GRACE dS/dt. Both solutions are very close to GRACE. However, in the "optimised storage" plots (right), only our sequential method fits low frequency GRACE storage changes. Figure 5 in the manuscript brings out this low frequency information, including from other products, that would not be seen if we only compare monthly P, Q and E estimates.

[Figure]

[Figure]

The main objective is a methodological advancement about how to use both short and long-term GRACE data to adjust/optimise the monthly fluxes to achieve a GRACE consistent long-term budget. In the manuscript we can try to emphasise more clearly that a timeseries of monthly fluxes is not adequate to identify long term consistency hence the storage figures that are perhaps less commonly shown. Would that satisfy your concerns?

**A second major, but not difficult to solve issue, is that I find the authors somewhat sloppy regarding equations and symbology. Unfortunately, this set of guidelines has disappeared recently from the HESS manuscript preparations guidelines online: https://iahs.info/Publications-News/Otherpublications/Guidelines-for-the-use-of-units-symbols-and-equations-in-hydrology.do, but I personally still appreciate it if we all try to follow this as much as possible. Of serious concern are Eq. (1) and Eq. (5), which should obviously read dS/dt instead of dS as the fluxes are per unit of time. It would also be helpful if the equations would contain the dimensions, thus, e.g. length^3 per time [L3 T -1 ] for Eq. (1) so this becomes obvious. Even expressed per unit area [L T-1 ] would also be fine of course as long it clearly remains a flux and not a stock. Moreover, please use single italicized symbols, so something like Sfi instead of FIS, which makes it directly clear we are talking about a storage. I know many other papers invent funny acronyms as well, maybe it is even the rule rather than the exception, but in my opinion, it is simply not pleasant for any reader.**

All equations and symbology will be corrected to follow appropriate guidelines. We will also rename FIS to S_fi, we appreciate this suggestion to help make our manuscript easier to follow.

**A third major question is why the authors chose the data they chose and whether it matters for the main point they are trying to make. For precipitation and evaporation, many more observation based products exist, so did they select the 'best' according to some previous studies or did they just select 'good' data and does it not matter a lot whether it is really the 'best'. I hope the authors can explain. Moreover, the runoff data is even dependent on precipitation and evaporation from GSWP3, which is a bit of a vague product in terms of how it was constructed and I think it may even rely partly on GPCP and FLUXCOM, making the estimates of P, E and Q not completely independent. Moreover, I fail to see why spatially varying runoff is necessary at all, as on the basin scale, the actual river discharge measurements at the river mouth would suffice for which, for example, the GSIM archive (Do et al., 2018) could have also been used**.

As our main aim was to present a methodological advancement. We chose what should be good data and we do note that many previous studies of this type produce ensemble products because there is no such thing as the "best data". The choices of datasets were not critical to the point we were trying to make. Although we have only used single data products, uncertainties we applied are based on previous studies that have used multiple products to estimate uncertainties. We chose to use EO based datasets where possible due to the nature of our funding, which led to the decision to use the GPCP product. We also chose to use global gridded products as this ensures uniformities of uncertainties across all basins and it was a future ambition to move towards a gridded version of our own product.

For Runoff the GRUN product is the only available global gridded runoff dataset we are aware of; it uses GSIM observations as input and has been validated against many river flow datasets using monthly river discharge from the GRDC, see Figure 3 from Ghiggi et al., (2019). It is correct that the runoff data is not completely independent from the precipitation dataset, and this is explicitly discussed in lines 500 - 509 in the manuscript. This has also been noted in other papers however we go further in explaining the likely qualitative impact on results, which is however quite small for the cases shown.

Would it address your concerns if we included a discussion along these lines in the manuscript?

References

Ghiggi, G., Humphrey, V., Seneviratne, S. I., and Gudmundsson, L.: GRUN: an observation-based global gridded runoff dataset from 1902 to 2014, Earth Syst. Sci. Data, 11, 1655–1674, https://doi.org/10.5194/essd-11-1655-2019, 2019.

---

## Author Comment (AC2)

Response to anonymous referee 1

**General comment**

**Overall, I believe this is an interesting work. The authors studied the water and energy budget in several larger rivers on both short and long-time scales. The manuscript could be accepted after major revisions.**

Thanks for your helpful feedback, we really appreciate all your comments and have tried our best to address each of your concerns.

**1.Please label the river basins in the Figure 1.**

We have produced an updated version of Figure 1 (below). This includes a key to label the basins.

[Figure]

| 1. Amazon | 5. Amur | 9. Lena | 13. Mackenzie | 17. St. Lawrence |
|---|---|---|---|---|
| 2. Nile | 6. Parana | 10. Niger | 14. Ganges | 18. Indus |
| 3. Congo | 7. Yenisei | 11.Zambezi | 15. Chari | 19. Syr-Darya |
| 4. Mississippi | 8. Ob | 12. Chang Jiang | 16. Volga | 20. Huang He |

Colours are used to make the black labels easier to read. The colour scale is associated with basin size.

**2. The boundary of the Amur River is not correct, which would make following results not right. Please check different maps to use the correct boundary.**

We used a mask taken from GFZ Terrestrial Water Storage product (Boergens et al., 2020).  However, we agree the Amur basin was odd compared to other maps, e.g., Hydrobasin https://www.hydrosheds.org/products/hydrobasins Lehner, B., Grill G. (2013).  We have contacted GFZ authors to enquire.

[Figure]

However, we have now created a new mask for the Amur basin based on the Hydrobasins and reproduced results for the Amur. Results have only some slight differences and the new manuscript will be updated based on this new mask.  The Amur is updated in Figures 4 and 8, values in Table 2, and Appendix Figures A and B. Discussion is also edited where necessary.

Figure 1 above now includes the new Amur boundary. We compared the boundary for all other basins against Hydrobasin and all others seem to be ok.

New Amur mask:

[Figure]

The black outline comes from converting the hydrobasin coordinates onto 0.5 degree grids to match the other data used.

**3. Please use appropriate font size and keep consistent in each figure. The font is too small to be readable.**

I have reproduced all figures with larger font size 13 for axis labels and legends, and font size 12 for axis values. Previous font size used was python default size of 10.

**4. Lines 374-381. It seems these are methodology, and should not be placed in the results.**

Relevant parts have been moved to methodology section. However, some of these lines are describing other products that are shown in Figure 5, and we believe them relevant in results section to explain this figure.

**5. Lines 419-427. This part is not well written. Each paragraph has only two or three lines. Please rearrange the text.**

These lines have been rewritten to improve the quality of writing, and the paragraphs have been restructured to make sure they flow better.

**6. Line 471. References are needed to support your statement.**

We have adjusted the statement to "In previous budget studies longer-timescale constraints on the water budget have often not been applied  Abolafia-Rosenzweig et al., 2021; Hobeichi et al., 2020)", and added references which are examples of studies where no long-term constraints have been used in optimisation.

**7. When talking about optimization, we always cannot forget some popular optimization algorithms, such as SCE, DDS, GA, etc. What are the differences between your method and these popular ones?**
Our optimisation involves linear budget equations only and therefore always has a unique monthly solution, so results should not depend on optimisation algorithm. The difference in our optimisation method comes from the constraints, which could also be embedded in other optimisation algorithms. We will try to be clearer about what we mean by optimisation in manuscript.

**8. When reading paper, we always want to see the differences between your study and previous ones. Lines 457-470 stressed the similarities but not the differences. Please dig a little bit more to show the differences.**

The similarities referred to here are only to emphasise that adjustments are of a similar size to previous budget closure studies (i.e. within Obs errors) but our results achieve an improved long term consistency of water storage changes with GRACE, wherein lies the difference in our results. We will re-emphasise this in text.

**9. From the conclusion, I can see the main contribution from your study is that you introduced a sequential optimisation approach. Other than this, is there any new findings that different from other studies? There are a lot of optimization method can do the similar job. I want to see new findings that can advance our understanding of the hydrological processes.**

The advance is in fully utilising the GRACE water storage observations to constrain regional water fluxes from monthly to decadal timescales. Previous optimisation studies have only used GRACE on a monthly timescale and have failed to match low frequency storage variations which are important to understand through hydrological modelling. By accounting for longer timescales, it enables us to use more information to provide stronger constraints than these other studies. These more consistent optimised products can then be more useful for model comparison studies for example. See also first response to reviewer 2.

**10. A better judgment of the selections of the river basins should be given. Is it because the observation data in these rivers are better than others? Or other reasons. Some important river basins, such as the Mekong River, are not selected. No rivers in Western Europe are selected. I don't mean you have to select all the rivers, but an appropriate reason should be given.**

The basins selected capture a range of imbalances in their observed budgets from the initial data, including basins with strong interannual variability, basins from a variety of latitudes, and basins that have other optimised flux products already in the literature. We are also restricted to larger basins, preferably with simple basin boundaries as these will have smaller GRACE storage errors as described in Weise et al. (2016). A more detailed discussion will be added into section 2.7 to give a clearer justification of the selection of basins. Note that a larger set of basins are included in the appendix figures.

References
Boergens, E., Dobslaw, H., Dill, R., Thomas, M., Dahle, C., Murböck, M., Flechtner, F. (2020): Modelling spatial covariances for terrestrial water storage variations verified with synthetic GRACE-FO data. International Journal on Geomathematics 11, 24, https://doi.org/10.1007/s13137-020-00160-0.

Lehner, B., Grill G. (2013). Global river hydrography and network routing: baseline data and new approaches to study the world's large river systems. Hydrological Processes, 27(15): 2171–2186. https://doi.org/10.1002/hyp.9740.

Wiese, D., Landerer, F. W., and Watkins, M. M. (2016). Quantifying and reducing leakage errors in the JPL RL05M GRACE mascon solution, Water Resources Research, 52, 7490–7502, https://doi.org/10.1002/2016WR019344

---

## Author Comment (AC3)

**Petch and co-authors study the water and energy balance of 20 large river basins. Given my own expertise, I will mainly comment on the water balance part. Using GRACE-derived terrestrial water storage changes and globally available observation-based product of precipitation, evaporation and runoff they first study the imbalance between these products.**

Thanks again for the review of our manuscript. The responses given for the general comments are as in our earlier response during discussion phase. We now add responses to the specific comments below.

**Then, I failed to understand what the usefulness of the "Our Optimised Storage" and associated figures 4 and 5 are. It seems quite circular to me that if you force it to match GRACE it matches GRACE better than other products. From a hydrologist perspective I would rather see the optimized P, Q and E compared to other products. Perhaps even an independent better regional precipitation or river discharge dataset for specific basins, to see whether the optimized fluxes match that better and which would then clearly demonstrate the strength of their method. I am not sure if this is going to require major changes to the paper, or just a clearer explanation of the objectives and results.**

'Our optimised storage' is the total water storage produced from integrating our optimised P, Q and E fluxes. The usefulness of this quantity is that it is more directly comparable to GRACE products, and is able to capture information that cannot easily be seen when looking at individual monthly flux components. Small imbalances in monthly fluxes which would be hard to validate with any single month of data, can still imply unrealistic long-term changes in a basins water storage e.g. associated with rising/lowering water tables. GRACE data does track these longer timescales. For this reason, a set of fluxes consistent with GRACE is more useful for evaluating modelling products for example, using which we may seek to attribute both short- and long-term water budget variations. Previous studies which have sought to develop observational flux products consistent with GRACE have failed to match low frequency variations which are important to understand through hydrological modelling.

Previous optimisation studies have only use GRACE on a monthly timescale. Using GRACE total water storage anomalies, rather than only monthly storage changes, enables us to use more information to provide stronger constraints than previous studies.

I have plotted the figure here to help explain further. This compares two optimisation outputs for the Mississippi basin, both of which use GRACE as input. In blue, we show our sequential optimisation approach described in the study, along with an optimisation that enforces monthly water balance but with no long-term constraint in green. The left figure shows monthly flux imbalances (P − E − Q) compared to the GRACE dS/dt. Both solutions are very close to GRACE. However, in the "optimised

storage" plots (right), only our sequential method fits low frequency GRACE storage changes. Figure 5

in the manuscript brings out this low frequency information, including from other products, that would not be seen if we only compare monthly P, Q and E estimates.

The main objective is a methods advancement on how to use both short and long-term GRACE data to adjust/optimise the monthly fluxes to achieve a GRACE consistent long-term budget.
In the manuscript we will emphasise that a timeseries of monthly fluxes is not adequate to identify long term consistency hence the accumulated storage figures are shown instead. Will that satisfy your concerns?

**A second major, but not difficult to solve issue, is that I find the authors somewhat sloppy regarding equations and symbology. Unfortunately, this set of guidelines has disappeared recently from the HESS manuscript preparations guidelines online: https://iahs.info/Publications-News/Otherpublications/Guidelines-for-the-use-of-units-symbols-and-equations-in-hydrology.do, but I personally still appreciate it if we all try to follow this as much as possible. Of serious concern are Eq. (1) and Eq. (5), which should obviously read dS/dt instead of dS as the fluxes are per unit of time. It would also be helpful if the equations would contain the dimensions, thus, e.g. length^3 per time [L3 T -1 ] for Eq. (1) so this becomes obvious. Even expressed per unit area [L T-1 ] would also be fine of course as long it clearly remains a flux and not a stock. Moreover, please use single italicized symbols, so something like Sfi instead of FIS, which makes it directly clear we are talking about a storage. I know many other papers invent funny acronyms as well, maybe it is even the rule rather than the exception, but in my opinion, it is simply not pleasant for any reader.**

All equations and symbology will be corrected to follow appropriate guidelines. We will also rename FIS to S_fi, we appreciate this suggestion to help make our manuscript easier to follow.

**A third major question is why the authors chose the data they chose and whether it matters for the main point they are trying to make. For precipitation and evaporation, many more observation based products exist, so did they select the 'best' according to some previous studies or did they just select 'good' data and does it not matter a lot whether it is really the 'best'. I hope the authors can explain. Moreover, the runoff data is even dependent on precipitation and evaporation from GSWP3, which is a bit of a vague product in terms of how it was constructed and I think it may even rely partly on GPCP and FLUXCOM, making the estimates of P, E and Q not completely independent. Moreover, I fail to see why spatially varying runoff is necessary at all, as on the basin scale, the actual river discharge measurements at the river mouth would suffice for which, for example, the GSIM archive (Do et al., 2018) could have also been used.**

As our main aim was to present a methodological advancement so we chose a "good data set" and we noted that many previous studies have used ensemble products because there is no "best data". The choices were not critical to the main points. Although we only used single data products, uncertainties are applied based on previous multi-product studies. We used EO based datasets where possible partly due to the nature of our funding, which led to the decision to use the GPCP product, however we also sought global gridded products as this ensures uniformities of uncertainties across all basins and it is a future ambition to move towards a gridded version of our own product.
For Runoff the GRUN product is the only available global gridded runoff dataset we are aware of; it uses GSIM observations as input and has been validated against many river flow datasets using monthly river discharge from the GRDC, see Figure 3 from Ghiggi et al., (2019). It is true that the runoff data is not completely independent from the precipitation dataset, and this is explicitly discussed in lines 500 - 509 in the manuscript. This has also been noted in other papers, however we go further in explaining the likely qualitative impact on results, which is still quite small for the cases shown.
Does this address your concerns if we include more discussion along these lines in the manuscript?

**Specific comments**
**L1-2: "improving climate and earth system models"** I would say 'validating' or 'assessing the capability of' which is to be done first before anything can be improved.

We agree with comment and this statement will be changed in manuscript to 'validating climate and earth system models.'
As well as validation, optimisation can also be useful prior to data assimilation of products into such models.

**L6: "the corresponding turbulent heat fluxes ranges between ± 10 W m−2 "** I suppose something should range between value x and value y, thus this sentence misses something.
This statement in abstract will be edited to give absolute maximum and minimum imbalances.

**L8: "This exposes mismatches in seasonal water storage"** Mismatches between what and what exactly?
We are referring to the mismatches between seasonal water storage seen by GRACE, and the seasonal water storage implied from raw P, Q and E observations. We will make this clear in the revised manuscript.

**L12: "The optimization also reduces formal uncertainties on individual flux components"** Sounds great, but I failed to clearly identify this result in the paper itself.
This is shown in Table 3 and discussed in section 5.2 'Uncertainty estimate' in the manuscript. In the revised manuscript we will more clearly emphasise this result.

**L14: "The FIS metrics"** What are 'the FIS metrics'?
Here, the FIS metric was referring to results gained when calculating the accumulated storage implied from optimised fluxes from other studies. This line in abstract will be rewritten to avoid using this phrase as it is not fully explained at this point.

**L23: "Water is a conservative quantity"** Technically speaking this statement is incorrect. Water is used by plants for photosynthesis and released by decomposition or fire. Probably it is an order of magnitude lower than the errors made in the products of P, E and Q, but not entirely negligible.
What we meant by this is that the mass of water will be conserved. We will change text to be clear on this.

**Table 1 "present" and general period statements** It is rather irrelevant whether e.g. GRUN is available until 'present' or that it starts in 1902, what matters is which years you used for the analysis.
The period column will be removed from Table 1, and we will instead state the period we downloaded each dataset for. "All datasets have been downloaded for the months between October 2001 and December 2013"

**L91 "evaporation"** I strongly support the use of evaporation over the ambiguous term evapotranspiration, see Miralles et al. (2020) for the arguments why that is, so perhaps you could simply use evaporation also elsewhere in the manuscript.
Thank you for pointing us to Miralles et al. (2020), we agree with this point and will use the term 'evaporation' instead of 'evapotranspiration' throughout the manuscript.

**Equation 5 The integral is between what and what? What does the to the power 0 between brackets mean? Is this equation supposed to present a time series? Then it would be clearer if Sfi,w(t) was explicitly written.**

This equation will be rewritten as following.

$$S_{fi,w}[t] = \int_0^t \left(\frac{dS}{dt}\right) dt + S[0] = \int_0^t (\mathbf{P} - \mathbf{E} - \mathbf{Q})dt + S[0].$$

We integrate between time 0 and an arbitrary time t. Where 0 represents the first month in our period. We have also added S[0] which is consistent with the step of initialising with GRACE storage from January 2002.

The superscript 0 was to indicate a first order correction hae been applied, this was only associated with the detrended flux inferred storage shown in figure 4 and is not relevant for the optimisation, so it has been removed from equation 5. Text has been changed to make this clear.

For the detrended flux inferred water storage, the first order correction was to match the mean of P-E-Q to the mean of dS/dt from GRACE. This quantity was calculated to help identify imbalances in the seasonal storage cycles between GRACE and the other flux observations. For the detrended flux inferred energy storage, the first order correction forced the long-term NET to be zero.

**Technical corrections**
 **L93: "Earths" Earth's**
This technical correction will be made.

**L101: "Land" land**
This technical correction will be made.

References
 Ghiggi, G., Humphrey, V., Seneviratne, S. I., and Gudmundsson, L.: GRUN: an observation-based global gridded runoff dataset from 1902 to 2014, Earth Syst. Sci. Data, 11, 1655–1674, https://doi.org/10.5194/essd-11-1655-2019, 2019.

---

## Author Comment (AC4)

**Response to reviewer 3**

**General comments**

**Petch et al presented a new method to derive monthly water and energy flow estimates consistent with observed water and energy budgets. The paper is generally well-written, and the topic is highly relevant to the HESS readership. However, I do have some concerns and suggestions:**

Thank you for taking the time to review our manuscript. We appreciate all your suggestions and have tried our best to respond to all your comments.

**The authors appear to claim that their optimization method works well by evaluating the results with GRACE - a product that was used in the optimization process. Please consider validation/evaluation with an independent product and/or different time periods.**

The aim of this paper, and others that have previously used GRACE to constrain the more usual hydrological fluxes, is to bring in a new source of information to bear on what are generally quite poorly known hydrological flux quantities. We have shown that our approach does this more successfully than previous attempts because it takes account of longer-term information contained within GRACE. We have brought information from different products together and made them consistent both with GRACE and with a closed water budget on all timescales. This is the measure of the evaluation. There is not really a more accurate independent data product that we could compare to. See also response 1 to reviewer 2.

**The authors aim to present better water and energy data and methods. For the effort to be impactful and meaningful, please share the data and the scripts (the scripts were shared, but I could not find any content in the readme file).**

Thanks for pointing out that the scripts were not being shared correctly, we will update the doi in the manuscript. They are available under the following link:

https://github.com/sammypetch/Water-and-energy-budgets

Data has been added to a folder named 'Optimised data', which can be found under the same link. This folder contains a .csv file for each of the five basins studied in more detail (Mississippi, Amazon, Congo, Huang He and Amur).  Data on additional basins can be made available upon request. The script is also available under the same link.

**Since the paper argues that the produced method constitutes an improvement upon current optimisation methods, it would be useful if the evaluation/comparison figures and results section could show a clearer distinction between comparisons with products that are "optimized" datasets and those that are not.**

Comparisons with "non-optimised" products is essentially a comparison with the input data. Such as Figure 4 which demonstrates the imbalances present initially. A comparison with other optimised products can be considered the comparison with the other dataset we show, in particular where we demonstrate that other products become inconsistent with GRACE water storage information on longer timescales.  We will make this clearer in the text where we discuss Figures 4 and 5, and when assessing the optimisation adjustments.

**Since the paper explicitly aims to improve optimization at all time scales (monthly, interannual, trend), it would be useful if the figures and results section could clearly and explicitly show the improvements at each of those time scales.**

We indeed aim to bring fluxes into GRACE consistency at all timescales. On a monthly timescale, consistency is shown through monthly budget closure, like other studies. In Figure 4 we aimed to show the impact of the optimisation and demonstrate the improvements from raw observations over different timescales. In Figure 4 (left) we can see strong divergences in the un-optimized storages, particularly the Mississippi and Huang He. For the Mississippi the un-optimized fluxes equate to 5.1 cm excess precipitation each month.

Improvements compared to other optimisation methods are seen primarily over long timescales. The other methods also use monthly constraints to balance the budget, and so, there should be no really "detectable improvement" at this timescale from other optimised products (also response 2 to referee 2). However, because of our longer timescale constraint we can see improvements over interannual to longer periods. In Figure 5 we show the storage implied from ours and other fluxes, to demonstrate this. For example, in the Huang He the CLASS product, despite optimisation, shows a difference to GRACE total water storage anomaly of around 9 cm by 2010, equivalent of a precipitation excess of ~1.28 cm every month. This may be small when considering only monthly fluxes, but over longer periods small imbalances can have a cumulative effect and cause a significant storage divergence.

**Specific comments**

**L53: "is these" should be "in these".**

This technical correction has been made.

**L106: Instead of "short and long time scales", please consider being more precise (e.g., monthly, interannual, long-term trend).**

We aimed to be consistent with GRACE on a monthly timescale, as well as in agreement with any interannual and long-term storage trend. We will mention these specific timescales in text.

**Other parts of the paper suggest that the aim is to both produce optimized estimates and an optimisation method/methodology. Please include all study aims in this "aim" paragraph.**

Yes, we aim to produce a new optimisation method as well as produce new estimates to demonstrate our method. Aims will be updated to state both.

**Introduction section: Please consider adding a table providing an overview of optimisation methods. The text already contains a literature review, but it is difficult to gain an overview. Since this paper proposes a methodological advancement, it would be useful to at a glance see in what way this paper presents an advancement.**

We hope that our new explanation of aims and objective (see comment 2 from reviewer 2) will help reader gain a clear overview of the advances we have made.

**Table 1: "present" is ambiguous, it would be clearer if you simply state the years that were downloaded for use in this study. Also make sure that the capitalisation of the headings are consistent. "Parameter" should be "Variable", I think. In addition, please consider adding a column**

**describing the dataset type (e.g., satellite, in-situ measurements etc). For GRACE, should the variable be "water storage anomaly"?**

The column containing the period of data availability has been removed and replaced with text to state the years downloaded for study. We have added a new column in the table to describe dataset type. All headings have been Capitalised and 'Parameter' has been changed to 'Variable'. And yes, this should be water storage anomaly! We have corrected this throughout the manuscript

**Methods section: Please consider adding an overview figure of the methodological steps. For variable symbols, please consider using single-letter symbols rather than multi-letter symbols.**

I appreciate these suggestions to make the methods easier to understand, and we have renamed FIS to S_fi. I have attempted to produce an overview figure of the methodological steps, see below. We have found this a useful figure, but not sure if it will be beneficial to the paper.

[Figure]

**Figure 4 (and elsewhere), please check - "total water storage" or "total water storage anomaly"?**

It indeed should read 'total water storage anomaly'. This has been corrected in all places in text necessary and changed on y axis labels in Figure 4 and 5.

**L350 First use of ITCZ, write out.**

ITCZ will be written out before first use.

**L461 Please consider providing the relative error in the unit of % for Amazon as well.**

The error here is 14 % expressed as a percentage of precipitation. We will include this in the manuscript.

**L468 Since the imbalances of the Amazon and Amur were explained by the lack of measurements, it seems odd that Congo is presented in this context as the basin with lowest imbalance without**

**further explanation. Between the lines, the text seems to imply that the lack of measurements is not as much an issue in the Congo, which is not true. If any, the lack of measurements is even a bigger issue in this region. Please consider a revision of the paragraph.**

We have now added text to explicitly say that the low imbalance in the Congo is not necessarily because of good coverage to avoid this implication. We have discussed possible explanations such as better observed rainfall due to TRMM, and that low imbalances can occur due to a cancelation of errors, but we cannot know for sure without further investigation. As the main aim of our paper was to present a methodological advancement, we do not go into more detail here.

**Sect 5.1. Consider moving relevant parts to the Methods.**

A subsection 'Goodness of fit' has been added to the methods and contains relevant parts of section 5.1.

**L551. Could the authors also share the optimized results?**

Results have now been shared, follow link in response to second comment.

**I could not find any content in the readme.md file beside a single row stating "Water-and-energy-budgets". I have attempted to view it both by downloading it and opening it using a text editor, and by previewing it on GitHub. Please check.**

This has been checked and should now be accessible from link given in second comment. I will make sure the doi in the manuscript works too.

---

## Author Response (AR1)

Point-by-point changes made in response to reviews

Referee 1 comments:

1. **Please label the river basins in the Figure 1.**

   Basins are now labelled in Figure 1.

2. **The boundary of the Amur River is not correct, which would make following results not right. Please check different maps to use the correct boundary.**

   A new mask has been made for the Amur based on Hydrobasin (Lehner, 2020). Figures 1 and 2 containing maps have been updated to include new mask. Figures 4 and 8 and appendix figures have been updated to include results obtained using the updated mask. Information in Table 2 and 3 have also been updated according to new mask.

3. **Please use appropriate font size and keep consistent in each figure. The font is too small to be readable.**

   All figures have been updated to use larger font size. Size 13 is used for axis labels and legends, and font size 12 for axis values. Previous font size used was python default size of 10.

4. **Lines 374-381. It seems these are methodology, and should not be placed in the results.**

   We have altered this text slightly but have mostly left these lines here as they describe the other products shown in Figure 6 and we believed they were relevant to explain this figure and did not include our own methods.

5. **Lines 419-427. This part is not well written. Each paragraph has only two or three lines. Please rearrange the text.**

   These paragraphs have been rewritten (now lines 455-465)

6. **Line 471. References are needed to support your statement.**

   Here we have referenced Abolafia-Rosenzweig et al., 2021 and Hobeichi et al., 2020 which are examples of studies where no long-term constraints have been used. Now line 509.

7. **When talking about optimization, we always cannot forget some popular optimization algorithms, such as SCE, DDS, GA, etc. What are the differences between your method and these popular ones?**

   We have added some text regarding choice of optimisation algorithm (lines 83-85)

8. **When reading paper, we always want to see the differences between your study and previous ones. Lines 457-470 stressed the similarities but not the differences. Please dig a little bit more to show the differences.**

   We have added text explaining what we are trying to emphasise through discussing similarities (lines 506-508).

9. **From the conclusion, I can see the main contribution from your study is that you introduced a sequential optimisation approach. Other than this, is there any new findings that different from other studies? There are a lot of optimization method can do the similar job. I want to see new findings that can advance our understanding of the hydrological processes.**

We have added text in conclusion to highlight the methodological advancement we have made in fulling utilising the GRACE storage data on multiple timescales also re-emphasised the advancement of our coupled energy budget (lines 563- 570).

10. **A better judgment of the selections of the river basins should be given. Is it because the observation data in these rivers are better than others? Or other reasons. Some important river basins, such as the Mekong River, are not selected. No rivers in Western Europe are selected. I don't mean you have to select all the rivers, but an appropriate reason should be given.**

Text has been added to discuss the selection of basins in section 2.7 (lines 229-232).

Referee 2 Comments:

1. **Then, I failed to understand what the usefulness of the "Our Optimised Storage" and associated figures 4 and 5 are. It seems quite circular to me that if you force it to match GRACE it matches GRACE better than other products. From a hydrologist perspective I would rather see the optimized P, Q and E compared to other products. Perhaps even an independent better regional precipitation or river discharge dataset for specific basins, to see whether the optimized fluxes match that better and which would then clearly demonstrate the strength of their method. I am not sure if this is going to require major changes to the paper, or just a clearer explanation of the objectives and results.**

Text has been added in sections 4.2 and 4.3 explaining why we have chosen to show "our optimised storage" and what the information that this quantity reveals.

2. **A second major, but not difficult to solve issue, is that I find the authors somewhat sloppy regarding equations and symbology. Unfortunately, this set of guidelines has disappeared recently from the HESS manuscript preparations guidelines online: https://iahs.info/Publications - News/Otherpublications/Guidelines-for-the-use-of-units-symbols-and-equations-in-hydrology.do, but I personally still appreciate it if we all try to follow this as much as possible. Of serious concern are Eq. (1) and Eq. (5), which should obviously read dS/dt instead of dS as the fluxes are per unit of time. It would also be helpful if the equations would contain the dimensions, thus, e.g. length^3 per time [L3 T -1 ] for Eq. (1) so this becomes obvious. Even expressed per unit area [L T-1 ] would also be fine of course as long it clearly remains a flux and not a stock. Moreover, please use single italicized symbols, so something like Sfi instead of FIS, which makes it directly clear we are talking about a storage. I know many other papers invent funny acronyms as well, maybe it is even the rule rather than the exception, but in my opinion, it is simply not pleasant for any reader.**

We have redefined 'FIS' to S_fi throughout text, and wrote dS/dt in places where dS was previously used. As well as some other alterations to the symbology.

3. **A third major question is why the authors chose the data they chose and whether it matters for the main point they are trying to make. For precipitation and evaporation, many more observation based products exist, so did they select the 'best' according to some previous studies or did they just select 'good' data and does it not matter a lot whether it is really the 'best'. I hope the authors can explain. Moreover, the runoff data is even dependent on precipitation and evaporation from GSWP3, which is a bit of a vague product in terms of how it was constructed and I think it may even rely partly on GPCP and FLUXCOM, making the estimates of P, E and Q not completely independent. Moreover, I fail to see why spatially varying runoff is necessary at all, as on the basin scale, the actual river discharge measurements at the river mouth would suffice for which, for example, the GSIM archive (Do et al., 2018) could have also been used.**

We have added text in Section 2 discussing the choice of dataset (lines 121-126).

4. **L1-2: "improving climate and earth system models" I would say 'validating' or 'assessing the capability of' which is to be done first before anything can be improved.**

This line has been changed to "assessing the capability of.." (lines 1-2).

5. **L6: "the corresponding turbulent heat fluxes ranges between ± 10 W m−2 " I suppose something should range between value x and value y, thus this sentence misses something.**

This has been replaced to say the imbalances ranges between 1 and 12 Wm-2 (lines 6-7).

6. **L8: "This exposes mismatches in seasonal water storage" Mismatches between what and what exactly?**

The new manuscript explicitly states that we mean the difference between GRACE and the storage suggested by the other flux observations (line 9).

7. **L14: "The FIS metrics" What are 'the FIS metrics'?**

This has phrase has been removed.

8. **L23: "Water is a conservative quantity" Technically speaking this statement is incorrect. Water is used by plants for photosynthesis and released by decomposition or fire. Probably it is an order of magnitude lower than the errors made in the products of P, E and Q, but not entirely negligible.**

This phrase has been replaced to say that the mass of water will remain constant (line 24).

9. **Table 1 "present" and general period statements It is rather irrelevant whether e.g. GRUN is available until 'present' or that it starts in 1902, what matters is which years you used for the analysis.**
The time resolution column has been removed from Table 1 and text has been added to state the years we downloaded the data for the study.

10. **L91 "evaporation" I strongly support the use of evaporation over the ambiguous term evapotranspiration, see Miralles et al. (2020) for the arguments why that is, so perhaps you could simply use evaporation also elsewhere in the manuscript.**

    The word 'evaporation' has been used throughout text now instead of 'evapotranspiration'.

11. **Equation 5 The integral is between what and what? What does the to the power 0 between brackets mean? Is this equation supposed to present a time series? Then it would be clearer if Sfi,w(t) was explicitly written.**

    Equation 5 has been rewritten. The power 0 is no longer used and limits have been added to the integrals.

12. **Technical corrections L93: "Earths" Earth's**
    This correction has been made (line 97).

13. **L101: "Land" land**
    This correction has been made (line 104).

Referee 3 comments:

1. **The authors appear to claim that their optimization method works well by evaluating the results with GRACE - a product that was used in the optimization process. Please consider validation/evaluation with an independent product and/or different time periods.**

    In section 4.2 we now explain the purpose of comparing results to GRACE (e.g. line 393-394).

2. **The authors aim to present better water and energy data and methods. For the effort to be impactful and meaningful, please share the data and the scripts (the scripts were shared, but I could not find any content in the readme file).**

    The data has been shared under new doi referenced in the revised manuscript.

3. **Since the paper argues that the produced method constitutes an improvement upon current optimisation methods, it would be useful if the evaluation/comparison figures and results section could show a clearer distinction between comparisons with products that are "optimized" datasets and those that are not.**

    We have now stated in text in section 4.3 that the comparison made in this section is with optimised products. It now reads "Each of these three products contains optimised estimates.."

4. **Since the paper explicitly aims to improve optimization at all time scales (monthly, interannual, trend), it would be useful if the figures and results section could clearly and explicitly show the improvements at each of those time scales.**

    Text has been added to section 4.2 and 4.3 to highlight how Figures 4 and 5 show improvements over multiple timescales (e.g. lines 395-401).

5. **L53: "is these" should be "in these".**

   This correction has been made (line 54).

6. **L106: Instead of "short and long time scales", please consider being more precise (e.g., monthly, interannual, long-term trend).**

   This line has been updated to include specific timescales (lines 112-113).

7. **Other parts of the paper suggest that the aim is to both produce optimized estimates and an optimisation method/methodology. Please include all study aims in this "aim" paragraph.**

   This aims paragraph has been updated to include all aims (line 109).

8. **Table 1: "present" is ambiguous, it would be clearer if you simply state the years that were downloaded for use in this study. Also make sure that the capitalisation of the headings are consistent. "Parameter" should be "Variable", I think. In addition, please consider adding a column describing the dataset type (e.g., satellite, in-situ measurements etc). For GRACE, should the variable be "water storage anomaly"?**

   The time resolution column has been removed from table 1 and the years of downloaded have been stated instead. A new column has been added to table 1 with heading 'Dataset type', 'Parameter' has been changed to 'Variable' and heading have been capitalised.

9. **Methods section: Please consider adding an overview figure of the methodological steps. For variable symbols, please consider using single-letter symbols rather than multi-letter symbols.**

   An overview figure has been added and the flux-inferred storage has been renamed to $S\_fi$ throughout manuscript.

10. **Figure 4 (and elsewhere), please check - "total water storage" or "total water storage anomaly"?**

    'Total water storage' has been replaced with 'Total water storage anomaly' through out text and in all figures necessary.

11. **L350 First use of ITCZ, write out.**

    Intertropical convergence zone has been written out before first use (line 370).

12. **L461 Please consider providing the relative error in the unit of % for Amazon as well.**

    This value has been added as a percentage of precipitation (line 500).

13. **L468 Since the imbalances of the Amazon and Amur were explained by the lack of measurements, it seems odd that Congo is presented in this context as the basin with lowest imbalance without further explanation. Between the lines, the text seems to imply that the lack of measurements is not as much an issue in the Congo, which is not true. If**

**any, the lack of measurements is even a bigger issue in this region. Please consider a revision of the paragraph.**

Text has been added to explicitly state that low imbalance is not necessarily due to good measurement coverage to avoid this implication (lines 503-505). This section also no longer discusses Amur imbalance.

14. **Sect 5.1. Consider moving relevant parts to the Methods.**

A new section named 'Goodness of fit' has been added in methods section 3.2.3. This contains relevant parts of section 5.1 (lines 335-339).

15. **L551. Could the authors also share the optimized results?**

Optimised results have now been shared and are available under new doi referenced in the revised manuscript. The text now also states that data on additional basins can be made available upon request.

16. **I could not find any content in the readme.md file beside a single row stating "Water-and-energy- budgets". I have attempted to view it both by downloading it and opening it using a text editor, and by previewing it on GitHub. Please check.**

A new doi has been included where the script and results should be accessible.

---

## Author Response (AR2)

Author Response

**Editor comment:**
**I can see that you have done a lot to reply to the concerns of the referees. The referees concur with that, but one issue remains. Referee #1 raised the issue of circularity: If you force the model on GRACE, then it is to be expected that the model also fits GRACE data well. In other words, it looks as if you have calibrated on GRACE and also validated on GRACE. For final acceptance you have to make clear that this is not (entirely) the case. Only if you can explain this issue well in your final submission I shall accept the paper for publication.**

We apologise for any confusion caused with regard to the point about validation. We are not using GRACE here to validate our model results. GRACE provides a valuable dataset which the other fluxes should be consistent with. Several previous papers have sought to develop a GRACE-consistent set of fluxes but do so incompletely. Our figures demonstrate clearly that previous products do not match long term GRACE variability within uncertainties, whereas our flux products do.

We have now added a sentence to the end of Sect. 4.1 describing how we are not attempting to validate our monthly fluxes.

Any "validation" of our products would require comparison with independent data that could be regarded as less uncertain that the datasets we have used. However monthly uncertainties are too large to distinguish between available products. We can see this in the figures below, which show the total water flux imbalance compared to GRACE from our and several other similar products. The shaded regions are the uncertainties taken from the CLASS product, although ours would look similar. These are the monthly values used to produce the total water storage anomalies in manuscript figure 6. Our additional constraint focuses on long timescale consistency, and so the benefits cannot be seen when looking at monthly timescales. But figure 6 still shows how the other products are inconsistent with GRACE over longer timescales, despite the monthly agreement within uncertainties shown here.

[Figure]

**Reviewer comments:**

**Part of the added text reads: "Overall, Fig. 5 highlights the initial large imbalances from combining different products and shows how these are removed and made consistent with GRACE on all timescales. On monthly timescales consistency is established through monthly budget closure, similar to other studies. The improvement compared to other optimisation methods however is seen over longer interannual timescales."**

**If I understand clearly there is still indeed some circularity in the reasoning, and I didn't get a real answer about why no validation to precipitation or river flow data was applied.**

**In any case the authors claim that their method and their 'our optimised storage' would outperform 'other methods'. Other methods are not applied, so there is no way for me to verify that claim.**

See response to point 1 above, including the comments about validation. Consistency with GRACE is a requirement which previous products do not properly satisfy whereas our approach does. We demonstrate this through Figure 6, where over longer timescales other products show inconsistencies with GRACE. We have altered text in section 4.2 and 4.3 to give clearer explanations.

**Moreover, the variable 'our optimised storage' is still not described in the methods section.**

We have now explained 'our optimised storage' in methods section 3.2.1.

**The point is that the water balance is not a physical law and now it's even somewhat worse by saying the volume is constant, which is even less true due to phase and density changes.**
**If the authors would just write: we assume water balance by:**
**dS/dt = P - E - Q**
**I would not argue with the authors as we all make that assumption, but it is what it is: an assumption to neglect the chemical reactions of water.**

We have changed this to say 'We assume water balance by: .. '